# A comprehensive molecular profiling approach reveals metabolic alterations that steer bone tissue regeneration

Julia Löffler[1,2,3], Anne Noom[1,2], Agnes Ellinghaus[1,2], Anke Dienelt[1,2], Stefan Kempa [3,4✉] & Georg N. Duda [1,2,4✉]

Bone regeneration after fracture is a complex process with high and dynamic energy demands. The impact of metabolism on bone healing progression and outcome, however, is so far understudied. Our comprehensive molecular profiling reveals that central metabolic pathways, such as glycolysis and the citric acid cycle, are differentially activated between rats with successful or compromised bone regeneration (young versus aged female Sprague-Dawley rats) early in the inflammatory phase of bone healing. We also found that the citric acid cycle intermediate succinate mediates individual cellular responses and plays a central role in successful bone healing. Succinate induces IL-1β in macrophages, enhances vessel formation, increases mesenchymal stromal cell migration, and potentiates osteogenic differentiation and matrix formation in vitro. Taken together, metabolites—here particularly succinate—are shown to play central roles as signaling molecules during the onset of healing and in steering bone tissue regeneration.

[1] Julius Wolff Institute (JWI), Berlin Institute of Health at Charité—Universitätsmedizin Berlin, 13353 Berlin, Germany. [2] BIH Center for Regenerative Therapies (BCRT), Berlin Institute of Health at Charité—Universitätsmedizin Berlin, 13353 Berlin, Germany. [3] Berlin Institute for Medical Systems Biology, Max-Delbrück-Center for Molecular Medicine, 10115 Berlin, Germany. [4]These authors contributed equally: Stefan Kempa, Georg N. Duda. ✉email: stefan.kempa@mdc-berlin.de; georg.duda@charite.de

In contrast to most tissues, bone has the unique capacity of scar-free healing after injury or fracture. Bone regeneration consists of well-balanced cellular and molecular pathways and cascades essential to re-establish tissue integrity and function. However, fracture non-unions, which occur in 10–15% of patients, put a significant burden on the individual patient and the health care systems[1,2]. Understanding of the underlying pathological mechanisms leading to delayed healing progression or non-union is key to improve treatment or even preventive care options.

Successful bone regeneration, results from a fine-balanced interplay of anabolic and catabolic processes, including inflammatory and angiogenic signaling, matrix formation and tissue remodeling[3]. These processes steering callus formation and fracture repair are driven by immune cells, endothelial cells, fibroblasts and stromal cells, all of which arrive early after injury in the fracture hematoma[3–5]. Beside immune cell functionality such as specific cytokine production also cell activation and proliferation are driven by metabolic pathways, which provide e.g., lipids as membrane building blocks and nucleotides for DNA replication[6–8]. This concept also applies to fibroblast or mesenchymal stromal cell proliferation and differentiation, which are critical processes during fracture repair[9,10]. It is without question that sufficient nutritional supply is central to effective healing, although the impact of this highly dynamic and time sensitive energetic demand has been only scarcely considered in regenerative research so far[11].

Considering the dynamic conditions during bone repair, comorbidities, such as diabetes, hormonal disorders and advanced age put additional stress on the already challenging metabolic setting and can impair bone healing[12,13]. Locally altered or limited nutrient availability may cause metabolic shortcomings in the callus tissue and result in different cell functionality and phenotypes—a concept long known for tumor cell transformation[14,15].

Since metabolic intermediates are not only relevant as biosynthetic precursors but can further act as ligands of specific receptors (e.g., G-Protein-coupled receptors)[16,17], cells engaging altered metabolic profiles can subsequently lead to changes in their communication. Fine-tuned communication and collaboration are the basis of successful re-organization and regeneration of an organ. Variation in this fine-tuned communicative interplay may consequence in disturbances or failure of bone regeneration and result in scar tissue formation, as frequently found in impaired revascularization or under prolonged or excessive inflammation[18].

So far, little attention has been paid to the local metabolic environment following a bone fracture incident. We hypothesize that the local metabolic environment is an essential driver of efficient bone tissue repair and metabolic alterations, especially during early bone healing. To prove this hypothesis, we used a novel angle by analyzing metabolic profiles and potential communication signals of local bone regeneration. To allow identification of differentiating signals between successful and compromised bone regeneration, we chose to pursue a comparative approach in early bone healing using two distinct experimental groups. This was done by utilizing a well-characterized animal model of successful versus biologically compromised fracture healing in female Sprague Dawley rats[19–21].

## Results

### Biologically compromised bone healing shows altered metabolic and immune responses at protein level compared to successful healing. We employed a model of biologically

compromised bone healing that manifests retarded bone formation in aged female Sprague-Dawley rats compared to young animals after receiving a 2 mm femoral osteotomy, as published[19–25] (Fig. 1 a, b, Supplementary Data 1). This model allows for the comparison of successful endogenous bone healing (young rats) to biologically compromised healing (aged ex-breeder rats, with a minimum litter of three), without any intervention to manipulate healing outcome. We started by validating the applicability of our model for early fracture healing time points (day 3, day 7, day 14) by histological, and gene expression analysis (Fig. 1c). Analysis confirmed progressed healing in young animals by faster callus progression (Fig. 2a, b) and higher levels of tissue mineralization (Fig. 2c, d, Supplementary Data 2). Lastly, fracture callus tissue from young rats showed increased expression levels of the osteogenic genes collagen type I alpha 2 chain (Col1a2) at day 3 and 7 and secreted phosphoprotein 1(Spp1) (Fig. 2e, Supplementary Data 2) at day 14.

To validate our hypothesis of the local metabolic environment being a driver of effective bone tissue repair, we performed untargeted proteomic screening (LC-MS/MS) of fracture hematoma/callus tissue sampled at days 3, 7, and 14 after femoral osteotomy from successful healing and biologically compromised healing, respectively.

To quantify protein levels using untargeted LC-MS proteomics, homogenized whole hematoma /callus tissue was used, as described in the method section under paragraph 4.3 "Animal sacrifice and sample harvest". The schematic workflow on sampling and down-stream analysis is depicted in Fig. 1c.

To identify up- and down- regulated proteins between groups of successful and compromised bone healing groups, fold changes of normalized label-free quantities (LFQ) intensities were calculated (compromised healing: successful healing). Subsequently, gene ontology (GO) term and KEGG pathway enrichment analyses were performed to identify differentially regulated proteins and protein cluster between successful and compromised healing. The strongest resulting regulation was seen in KEGG pathways associated with extracellular matrix and migration, metabolism, and immune responses (Fig. 3a; Supplementary Data 3, 16, 17). In support of our hypothesis, regulation of metabolic proteins between successful and compromised bone healing was among the most differentially regulated protein cluster. In particular, clusters related to sugar metabolism, amino acid metabolism and oxidative metabolism were decreased in the group of compromised healing (Fig. 3a; Supplementary Data 3; 16). By contrast, proteins related to immune response and secondary metabolic pathways were increased in compromised bone healing when compared to successful healing (Fig. 3b; Supplementary Data 3; 17). The results on protein level were indicative of a prolonged inflammatory phase in animals showing compromised healing, complementing the histological findings of an extended initial blood clot presence in the osteotomy gap (Fig. 2c).

By proceeding with a detailed analysis of the identified protein cluster, we found inflammatory and innate immunity-related proteins, like cathepsin G (CTSG), complement component 6 (C6), or fibrinogen gamma chain (FGG) particularly up-regulated in compromised bone healing tissue shortly after osteotomy (from day 3 onwards), lasting until day 14 (e.g., cathepsin A (CTSA), lysosome-associated membrane protein 2 (LAMP2), Fig. 3c). Levels of glucose and TCA cycle metabolism-associated proteins (e.g., phosphofructokinase (PFKM), isocitrate dehydrogenase 1 (IDH1)) started to increase in samples showing successful healing at day 7 and were highly increased at day 14 (e.g., isocitrate dehydrogenase 2 (IDH2), citrate synthase (CS), Fig. 4a, Supplementary Data 4). The most pronounced differences between successful and compromised bone healing were detected in proteins related to oxidative metabolism/phosphorylation

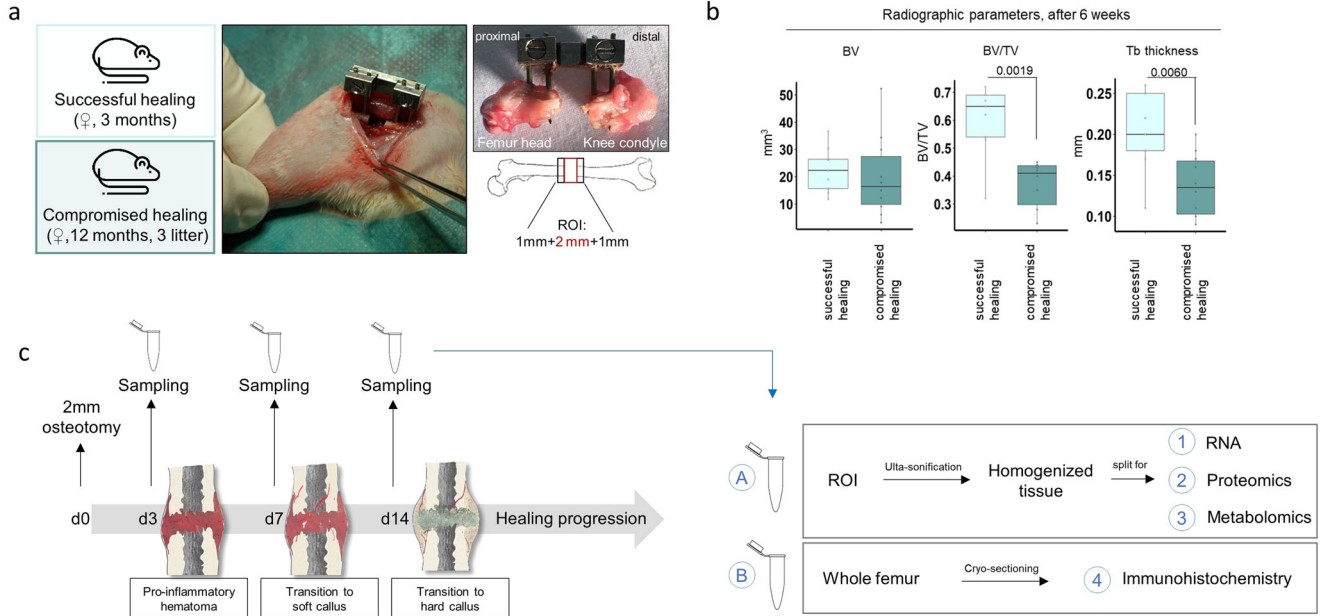

**Fig. 1 Schematic overview of applied in vivo fracture healing model and experimental workflow. a** Applied surgical procedure introducing a 2 mm standardized osteotomy gap in the left femur of young rats, aged 3 months, showing normal bone healing are compared to aged rats (12 months) that had a minimum litter of three, showing biologically compromised bone healing. **b** Newly formed bone has different properties in successful healing compared to compromised healing. Successful healing showed higher bone volume ratio and trabecular thickness, while bone volume did not change in comparison to compromised healing after 6 weeks. n = 9–10 individual biological replicates, Mann-Whitney U test[19,21,22] Central image: external fixator placed on the left femur, right image: dissected femur, where the region of interest (ROI: 2 mm hematoma + 1 mm proximal and distal to fracture gap) has been removed. All bones were placed in the same orientation for histological analysis, with the femur head (proximal) on the left side and the knee condyle (distal) on the right side of the image. BV - bone volume, TV- total volume, Tb – trabecular. Rat icon obtained from: Flaticon.com, artist: Nhor Phai. **c** Samples were harvested at day 3, 7 and 14 after osteotomy and subjected to the different downstream analysis, ROI – region of interest.

(OXPHOS) and highly upregulated in successful healing at day 14 after osteotomy e.g., NADH dehydrogenase [ubiquinone] 1 beta subcomplex subunit 9 (NDUF9b), ubiquinol-cytochrome c reductase, Rieske iron-sulfur polypeptide 1 (UQCRSF1, Supplementary Fig 1; Supplementary Data 11).

Gene expression analysis of inflammatory markers and metabolic enzymes were performed to validate findings on protein level. Expression of pro-inflammatory genes like the tumor necrosis factor-alpha (*TNF-alpha*) or nitric oxide synthase 2 (*Nos2*) was high in compromised healing samples at day 3. In contrast, expression of the anti-inflammatory cytokine interleukin-10 (*Il-10*) was higher in fracture samples from successful healing on days 3 and 7. Similarly, central carbon metabolic enzymes (hexokinase 2, *Hk2*, and succinate dehydrogenase beta, *Sdhb*) showed higher expression levels in successful healing samples during early healing time points, namely day 3 and day 7 (Fig. 4b, Supplementary Data 4).

**Checkpoints of central carbon metabolism show alterations between successful and compromised healing after 7 days**. To validate whether the observed regulations of metabolic pathways proteins are related to an altered metabolism, the second part of the homogenized hematoma/callus tissue specimen was subjected to GC-MS-based metabolic profiling. Metabolic intermediates of the central carbon metabolism (CCM) were identified using an analytical workflow as published by Kuich et al.[26]. While no significant differences in metabolic intermediates between the healing groups were detected at day 3 (Fig. 4a), differences in CCM metabolites at metabolic pathway checkpoints (e.g., 3-phosphoglyceric acid (3PG), lactate, and succinate) were detected (Fig. 5a, b, Supplementary Data 5) in day 7 samples. It is important to note, that glucose levels were comparable between

successful and compromised bone healing across all time points measured (Fig. 6a, Supplementary Data 6), suggesting no substrate limitations at the beginning of the glycolytic pathway. Lactate showed a 1.4-fold upregulation in samples from successful healing compared to compromised healing, while down-stream metabolites of the CCM, like pyruvate - which connects glycolysis and the tricarboxylic acid cycle (TCA cycle)- and citrate, an intermediate of TCA cycle, showed no differences in measurements.

Most interestingly, succinate levels were increased significantly at day 7 in samples from successful healing compared to compromised healing by 3-fold (Fig. 6a, Supplementary Data 6). Levels of the TCA cycle intermediates following succinate–fumarate and malate–were not altered between the bone healing groups (Fig. 6a).

At day 14 after osteotomy, no differences in lactate or succinate levels were detectable between the groups. However, pyruvate and citrate levels were reduced in samples from animals showing compromised healing (2.2-fold and 1.6-fold, respectively) compared to the group of successful bone healing (Fig. 6a). Similar tendencies were detected for the metabolites glutamate, α-ketoglutarate and protein levels of alpha-ketoglutarate dehydrogenase (OGDH), suggesting increased glycolytic and TCA cycle activity and TCA cycle replenishment from glutaminolysis in animals with a successful healing progression (Fig. 6a & Supplementary fig 2, Supplementary Data 6, 12).

Early regulation of the two CCM intermediates, succinate, and lactate, marked them as interesting candidates in regard to their roles during the bone healing process and progress. Both metabolic intermediates have already been discussed as mediators of immune cell responses in literature. Particularly, a study by Tannahill and colleagues, demonstrated significant effects of intracellular accumulation of succinate on interleukin 1-beta (*IL-

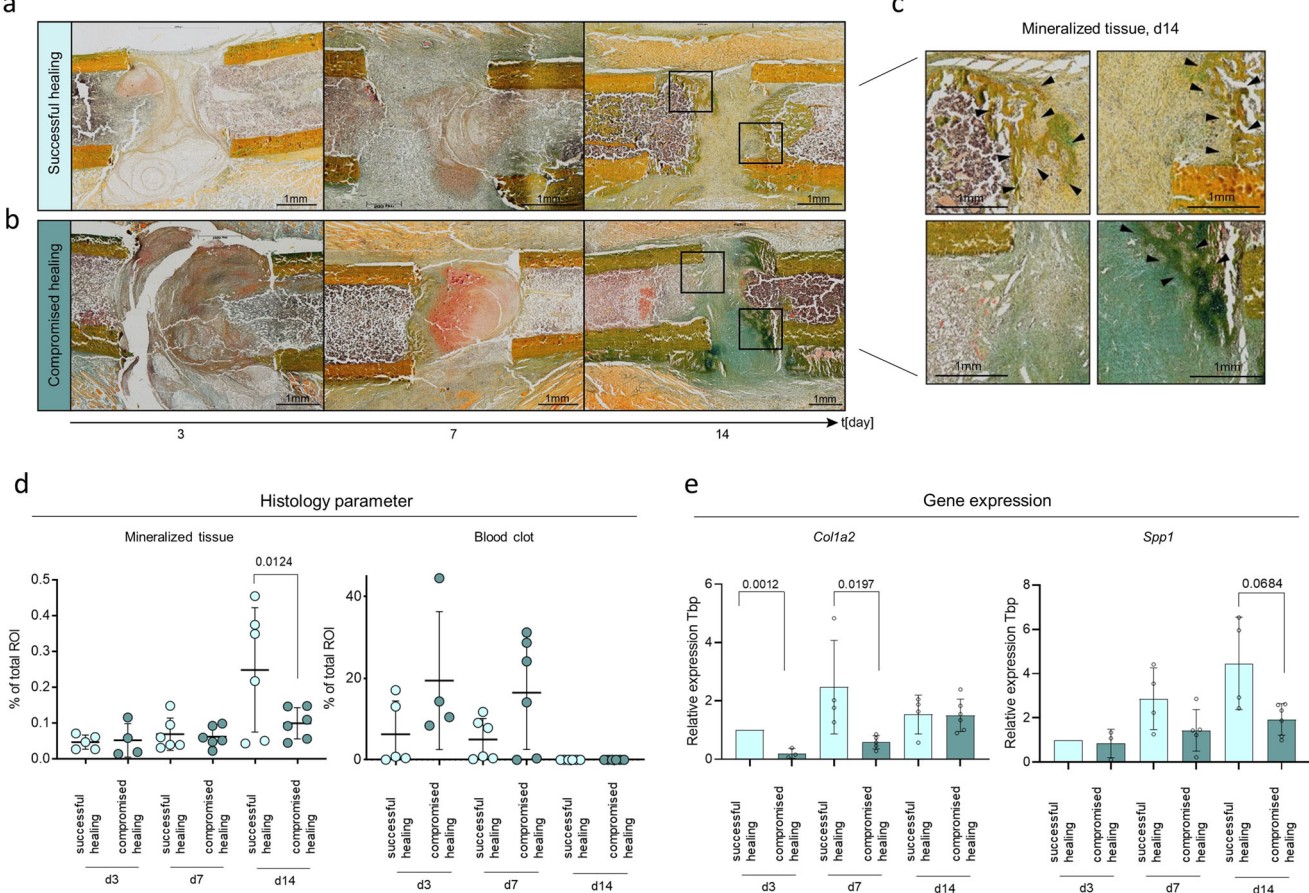

**Fig. 2 Validation of the in vivo model suitability for the early healing time-points: day 3, day 7, and day 14 by histological assessment and gene expression analysis of osteogenic markers. a** Tissue distribution (mineralized tissue, blot clot tissue) evaluated by Movat-Pentachrome staining. Day 3, 7, and 14 after osteotomy shown from left to right showing successful healing. Femurs are placed in the same orientation, with the proximal part on the left side and the distal part on the right side of the image. Black boxes indicate the magnified area displayed in Fig. 2c. **b** The panel shows an example for compromised healing, days 3, 7, and 14 after osteotomy arranged from left to right. Femurs are placed in the same orientation, with the proximal part on the left side and the distal part on the right side of the image. Black boxes indicate the magnified area displayed in Fig. 2c. **c** Magnification of light microscopic images from Figs. 2a and b. Arrows indicate newly mineralized tissue. Movat-Pentachrome staining: yellow color = mineralized tissue, red/brown = blood clot. Proximal part on the left side and the distal part on the right side of the image. **d** Tissue quantification of Movat-Pentachrome-stainings showed higher tissue mineralization in successful healing conditions at day d14. Compromised healing showed longer presence from initial blood clot tissue. $n = 6$ individual biological replicates per group and time point analyzed, One-way ANOVA. **e** Osteogenic factor, *Col1a2* (collagen type I alpha 2 chain) showed significantly higher expression at day 3 and day 7 and *Spp1* (osteopontin) at day 14 after osteotomy in successful healing, relative expression to *Tbp* (TATA-binding protein, housekeeping gene) and d3 young, $n = 3$–6 individual biological replicates per group and time point, One-way ANOVA. Mean±Standard deviation shown for all graphs.

*1β*) expression by hypoxia-inducible factor 1-alpha (*HIF-1α*) stabilization in macrophages[27]. Further studies showed that succinate has additional properties as an extracellular signaling molecule, mediated over its G-protein-coupled receptor, succinate receptor 1(SUCNR1)[16,28,29]. Interestingly, *SUCNR1* is highly expressed in bone marrow and blood cells, particularly MSCs, monocytes, and macrophages (Human protein atlas, Version 19.3)[30,31].

Gene expression analysis of the succinate receptor in fracture samples showed an increased expression in samples of successful healing at day 7 (Fig. 6b, Supplementary Data 6). We were interested whether altered cytokine expression of macrophages, e.g., by intracellular succinate accumulation, may influence cellular crosstalk and lead to healing cascade alterations. Metabolic analysis of serum samples collected at the time of animal sacrifice confirmed that there was no systemic regulation of succinate between healing time points in successful or compromised healing (Supplementary Fig 3, Supplementary Data 13). Unfortunately, localization and

cellular origin of the accumulated succinate was not possible in the applied setting, due to the usage of homogenized hematoma/callus tissue samples.

We, therefore, decided to focus our attention on the potential function of extracellular succinate as a signaling molecule during the process of bone healing. We could demonstrate a differential expression of macrophage markers between young and old animals in a previous study, showing not only higher monocyte-macrophage marker expression, like *Cd14* and *Cd68* in fracture tissue of successful healing but also increased anti-inflammatory M2 macrophage gene expression[23]. Additionally, angiogenic marker genes, like *Hif-1alpha* and others associated with overall vessel formation were increased in fractures from young animals at day 7. Further studies performed at our institute further confirmed that M2 macrophages appear in the fracture gap at around day 7 in the selected animal model[32]. We assumed that the observed difference in succinate levels between the successful and compromised healing animals could be related to macrophages and aimed to explore this potential crosstalk further.

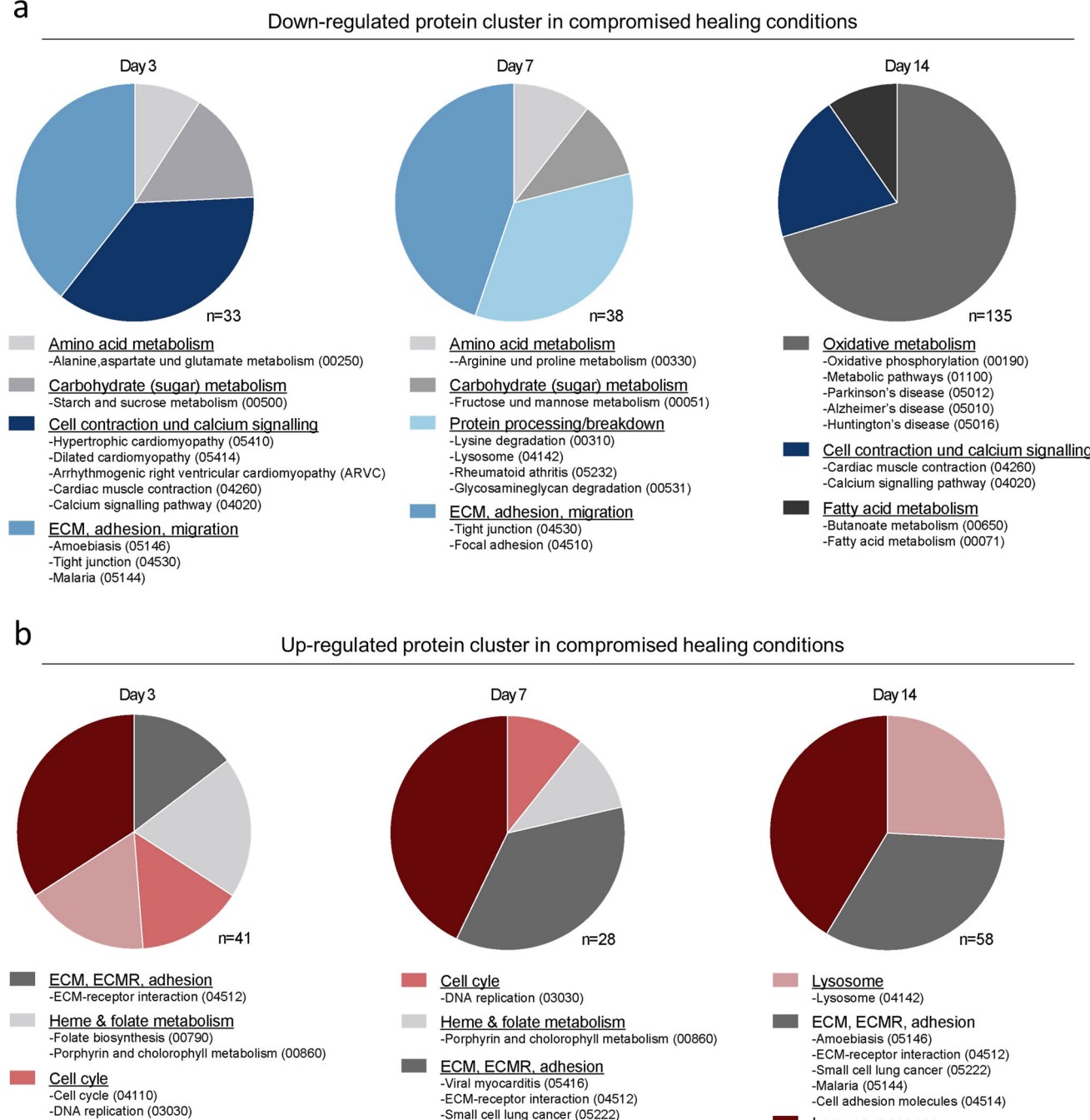

**Fig. 3 GO and KEGG enrichment analyses of the fracture proteome reveal distinct differences in metabolic-, immune- and matrix-related processes between successful and compromised bone healing in rats. a** Down-regulated pathways and processes in compromised bone healing tissue are related to amino acid and carbohydrate metabolism and extracellular matrix at day 3 and day 7, while oxidative metabolism processes are downregulated at day 14. Cell contraction and calcium signaling proteins are downregulated at day 3 and day 14 in particular. Total protein number (*n*) as identified by GO and KEGG enrichment analyses are indicated for each timepoint. **b** Protein cluster associated with immune responses and extracellular matrix—receptor interactions have been found up-regulated in compromised healing for all time points, while cell cycle proteins were upregulated at day 3 and 7 and lysosomal proteins increased at day 3 and day 14. Total protein number (*n*) as identified by GO and KEGG enrichment analyses are indicated for each time-point.

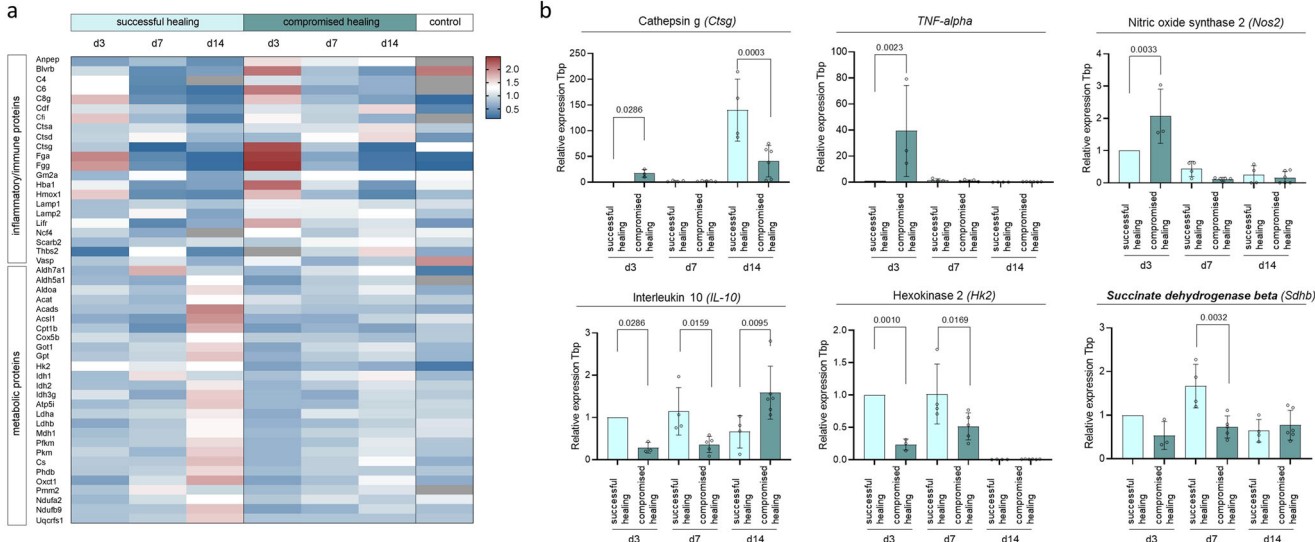

**Fig. 4 Expression of selected metabolic and inflammatory genes from fracture tissue validates proteomic findings of altered metabolism and inflammation between successful and compromised bone healing. a** Heatmap of proteins related to the immune response and metabolism highlighting specific profiles for successful and compromised healing. Day 3, in particular, showed increased levels of immune and inflammatory proteins in tissue of compromised healing fractures. Proteins from metabolic pathways showed an increasing trend in successful healing fracture tissue at day 7, and a strong upregulation compared to compromised healing at day 14. Shown are sum over mean normalized LFQ intensities, with following color coding, red= upregulated, white= unchanged between conditions, blue= downregulated, grey= no values. n = 3–5 biological replicates (BR) per group and time-point. **b** Gene expression analysis of selected inflammatory and metabolic markers validate findings from protein levels. Pro-inflammatory genes like *Ctgs, Tnf-alpha* or *Nos2*, showed significant higher expression in compromised healing samples. The anti-inflammatory gene *Il-10* and enzymes from central carbon metabolism (*Hk2, Sdhb*) showed higher expression in samples from successful healing. Relative expression to *Tbp* (TATA-binding protein, housekeeping gene) and d3 young, n = 3–6 individual biological replicates per group and time point, One-way ANOVA, mean±standard deviation.

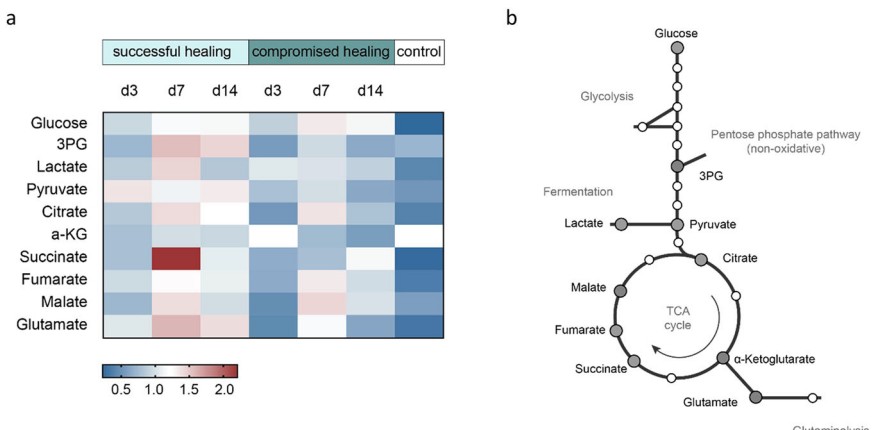

**Fig. 5 Untargeted metabolomics of rat fracture tissue from successful and compromised bone healing reveal differences in intermediates from the central carbon metabolism. a** Heatmap of selected central metabolites from the central carbon metabolism (glycolysis, TCA cycle). Shown are sum over mean normalized LFQ intensities, with following color coding, red= upregulated, white= unchanged between conditions, blue= downregulated, grey= no values. n = 3–5 biological replicates (BR) per group and time-point. **b** Schematic central metabolic pathway, metabolites that are shown in heatmap are marked by grey and enlarged circles.

We did so, by simulating a local exposure of succinate to cells relevant in bone healing, like macrophages, mesenchymal stromal cells (MSCs) or endothelial cells in vitro, aiming to mimic the microenvironment during successful fracture healing.

**Extracellular succinate induces transcriptional and functional changes in activated human macrophages and CD14+ cells in vitro.** First, effects of physiological concentrations of extracellular succinate (50 and 500 μM[33,34]) on monocytes/macrophages were analyzed. The shift from M1 to M2 macrophages is a strong indicator for the transformation from the pro- to the

anti-inflammatory phase in the early hematoma and a crucial step in successful endogenous bone healing[18,23]. Human monocytic cells (THP-1 cell line) were chosen to study the effect of extracellular succinate exposure (50 and 500 μM) on the differentiation of M1 and M2-like macrophages in vitro. Changes in transcription of M1- and M2-like macrophage markers and cytokines, upon extracellular stimulation with succinate, were analyzed (for differentiation experiments of monocytic cells into macrophage subsets please refer to Supplementary Fig 4, Supplementary Data 14). Addition of succinate to the cultures led to an increase of *IL-1β* expression in M1 versus M2 macrophages (Fig. 7a, Supplementary Data 7). IL-1β secretion was likewise

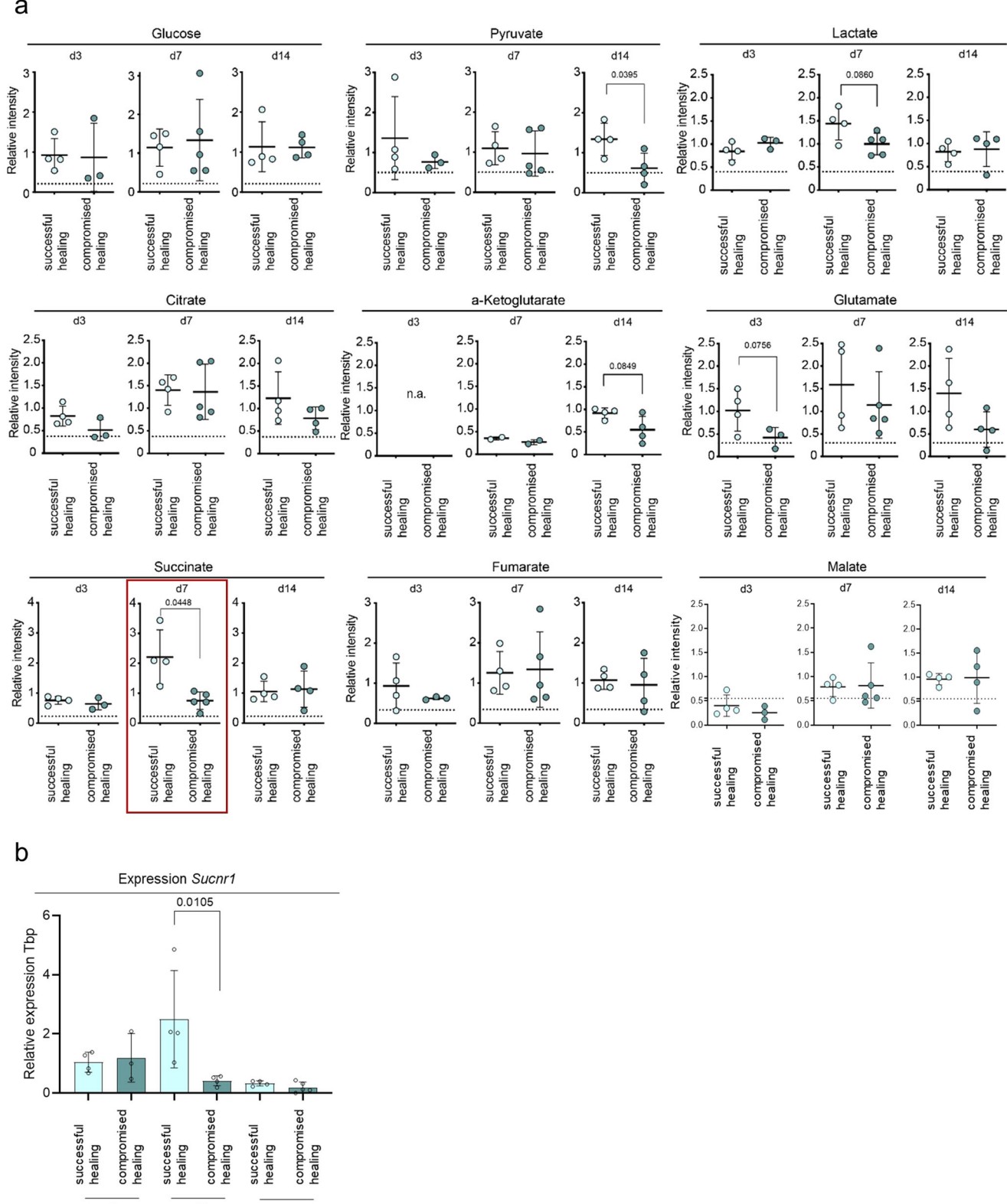

increased in M1 macrophages upon addition of succinate to the culture (Fig. 7b, Supplementary Data 7). When freshly isolated human cluster of differentiation 14 positive (CD14⁺) cells were activated with lipopolaysaccharide (LPS) and stimulated with 50 μM succinate, IL-1β secretion was also increased (Fig. 7c, Supplementary Data 7). These results complement previous studies from Tannahill and colleagues, which showed

that intracellular accumulation of succinate leads to *IL-1β* expression[27,35].

**Extracellular succinate enhances angiogenic and migratory processes of primary human cells in vitro**. As angiogenesis and wound closure are essential steps in bone healing and closely

**Fig. 6 Analysis of central carbon metabolites shows increased levels of succinate and expression of the succinate receptor gene at day 7 for successful healing fractures. a** Relative label-free qualification of selected metabolites from untargeted metabolite screening. No differences in relative glucose levels between successful and compromised healing identified at day 7 but an increased trend for lactate in successful healing. At day 14 a general increase of the central carbon metabolism (especially the TCA cycle) was detectable in successful bone healing, as seen in significantly increased levels of pyruvate and increased trends in citrate, glutamate, and α-ketoglutarate levels. Succinate showed significantly increased values at day 7 in successful bone healing compared to compromised healing, $n = 3$–5 individual biological replicates per group and timepoint, t-test, the dotted line represents control values of unfractured, contralateral bone. **b** Expression of the succinate receptor 1 (*Sucnr1*) gene is increased at day 7 in successful healing samples. Relative expression to *Tbp* (TATA-binding protein, housekeeping gene) and d3 young, $n = 3$–5 individual biological replicates per group and time point, One-way ANOVA. Mean±standard deviation is shown for all graphs.

linked to the switch from pro- to anti-inflammation, further experiments targeted the modulation of these two steps by extracellular succinate. Tannahill and colleagues showed that intracellular succinate induces mRNA expression and HIF-1α activity[27]. When we investigated in vitro tube formation using human HUVECs (hHUVECs), addition of succinate to cultures significantly enhanced tube length and network formation (Fig. 7d;7, Supplementary Data 7) under angiogenic culture conditions (+GF, +FCS, EGM bullet kit, Lonza). This was particularly seen when adding 50 μM of succinate. Remarkably, extracellular stimulation of succinate (50 μM) also led to increased tube formation in growth factor-free conditions (−GF, +FCS) (Fig. 7d, Supplementary Data 7), indicating that extracellular succinate can function as an angiogenic factor. This result complements and extends findings by Mu et al., which demonstrated that higher concentrations of succinate (200–800 μM) promoted the formation of tubular structures in vitro and in zebrafish[36].

Wound healing and migration of cellular progenitors into the fracture area is a vital part to bone regeneration[37]. Therefore we analyzed, how succinate affects the regenerative capability of primary human mesenchymal stromal cells (hMSCs), obtained from patients undergoing total joint replacement surgery.

Addition of succinate to a scratch/wound healing assay considerably enhanced the migration of primary human MSCs. Cells from three independent donors were investigated, all showing higher percentages of repopulated scratch area, when succinate was added to the culture media (15–20%, Fig. 7f, g, Supplementary Data 7). Monitoring hMSC migration over a time of 14 hours showed, that upon treatment with succinate an increased migratory rate was detected compared to control conditions without succinate addition. This effect was significant for treatments with 50 μM succinate at 15 hours post-scratch and onwards, also for treatments with 500 μM succinate from 21 hours post-scratch and onwards (Fig. 7h, Supplementary Data 7). As Ko and colleagues showed, this effect is most probably mediated by the succinate receptor SUCNR1[38].

**Osteogenic differentiation potential of primary human MSCs increased by extracellular stimulation with succinate in vitro.** Osteogenic differentiation using primary human hMSCs obtained from different donors undergoing total joint replacement surgery was performed to investigate the effect of extracellular succinate stimulation on bone formation. Differentiation was performed using a well-established protocol for a period of 18 days (details see Methods). Enzymatic activity of alkaline phosphatase (ALP), marker for early osteogenic differentiation, was significantly increased at day 7 in osteogenic cultures (OM), upon addition of succinate to the culture medium in all tested donors (Fig. 8a, Supplementary Data 8). Similarly, after 18 days of osteogenic differentiation cultures, matrix formation and mineralization were increased when succinate was added to the culture (Fig. 8b, d, Supplementary Data 8). To account for cell division, the cell number was determined by Hoechst staining and did not show any differences between the treatments (Fig. 8c, Supplementary Data 8).

Addition of succinate (500 μM) in vitro particularly induced strong mineralization, when the overall mineralization capacity appeared delayed or less pronounced, as suggested by observations from donor three (Fig. 8c, Supplementary Data 8).

**Combining effects of extracellular succinate and transient IL-1β enhance matrix mineralization during osteogenic differentiation of primary MCSs in vitro.** Considering that extracellular succinate induced IL-1β expression and secretion from LPS stimulated macrophages, addition of both, succinate and IL-1β, was investigated on osteogenic differentiation of human MSCs. While constant IL-1β (0.1 ng/ml, 10 ng/ml)[39] lead to a decreased osteogenic differentiation and premature cell contraction (Supplementary Fig 5, Supplementary Data 15), a transient addition of 0.1 ng/ml IL-1β (24 h) to hMSCs under osteogenic culture conditions, led to a significant increase of mineralized matrix formation after 18 days (Fig. 9 a, c, Supplementary Data 9). Additional supplementation with succinate significantly enhanced the formation of matrix even further in 2 out of 3 donors and by trend in 1 additional donor (Fig. 9a, c, Supplementary Data 9). Cell number showed no significant alterations between the treatment groups (Fig. 9b, Supplementary Data 9).

**Paracrine signals from macrophages stimulated by extracellular succinate generate osteogenic induction in the absence of further osteogenic stimuli in vitro.** To further investigate the paracrine crosstalk between succinate-stimulated macrophages and MSCs during bone healing progression, osteogenic differentiation assays were performed using conditioned media from M1- and M2-like macrophages that have been cultured and polarized with and without succinate (50 and 500 μM) addition to the culture medium. This culture media was harvested and given to osteogenic differentiation assays using primary human MSCs. The addition of media from M2-like macrophages in conditions with osteogenic additives led to a significant increase of matrix mineralization as determined by alizarin red staining. This effect was even more pronounced when succinate was added to the cultures during macrophage polarization and stimulation. Addition of conditioned media from M1-like macrophages however, showed decreased mineralization, again the effect was stronger, when succinate was added to the cultures during macrophage polarization (Fig. 10a, b, Supplementary Data 10).

Most astonishing was the effect observed in cultures without any osteogenic supplements, that usually served as control conditions and where expansion medium was used. Here addition of conditioned media from macrophage polarizations induced mineralization, independent of macrophage polarization (M1 or M2) as seen in Figs. 10a and b. In control conditions, when media from M2-like macrophages stimulated with 50 μM of succinate during polarization and differentiation was added, levels of mineralization showed the highest results.

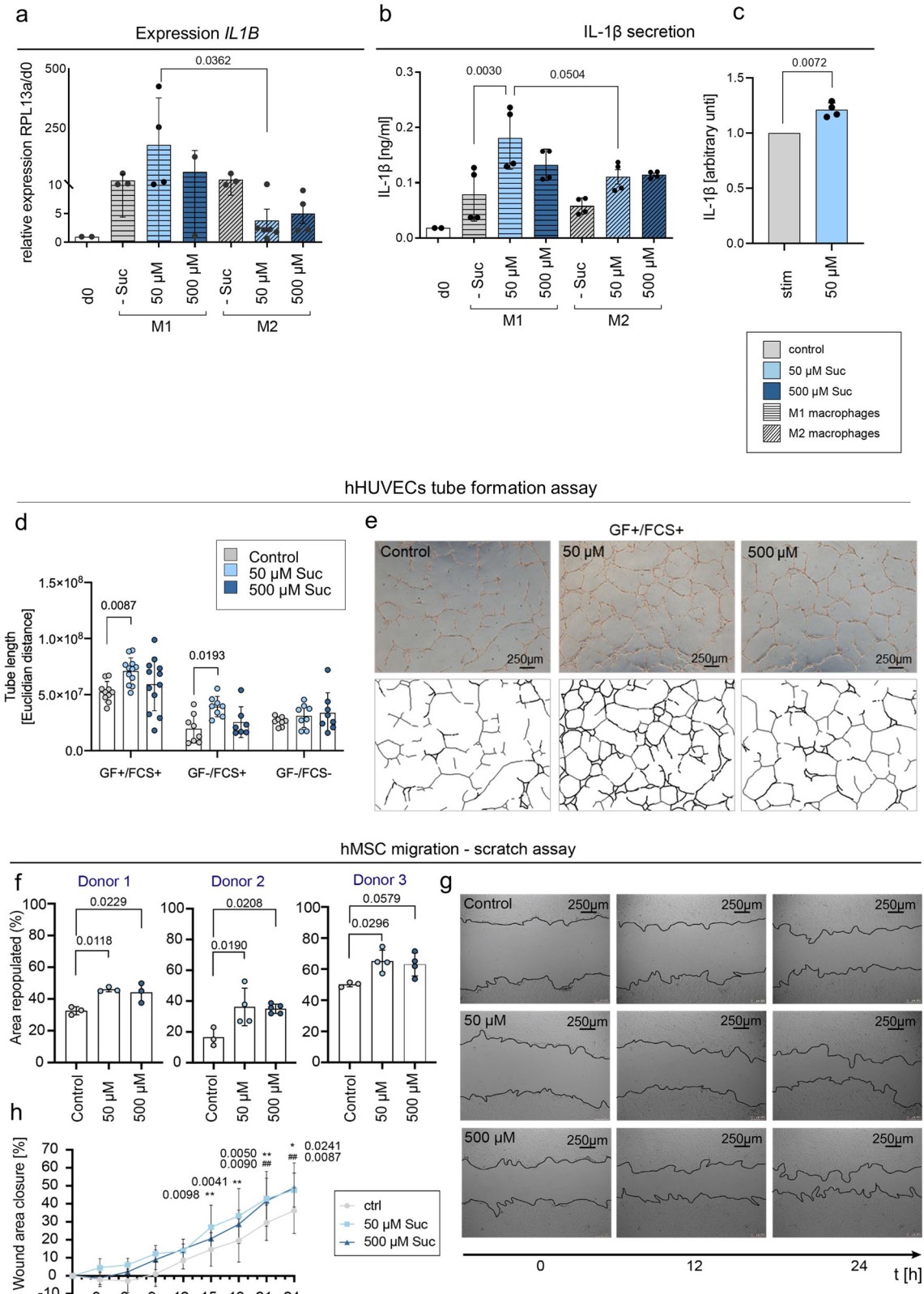

## Discussion

Bone is one of the few organs that can regenerate without scar formation. This capacity, though in principle well conserved until high ages, appears to be hindered or retarded in many aged individuals and is often linked to prolonged or excessive inflammation, instable fracture fixation or disturbed angiogenesis[18,23,32,40,41]. The fine-tuned and complex regenerative process is highly dependent on molecular and cellular communication and their interplay. How local nutrient supply affects cellular metabolism and functionality is important in understanding scar-free and successful regeneration. However, the role of metabolic cascades in early bone healing has not been intensively researched so far. Here we present, to the

**Fig. 7 Modulation effects of different cell populations by extracellular succinate application. a** Physiological concentrations of extracellular succinate (50 and 500 µM) upregulate the transcription of the IL-1β gene in activated THP-1 M1 macrophages, $n = 3$ experimental replicates, Relative expression to *RPL13A* and d0. One-way ANOVA, Suc – succinate. **b** IL-1 β secretion was significantly increased in M1 macrophages when stimulated with 50 µM succinate, $n = 3$ individual experimental replicates, One-way ANOVA, Suc - succinate. **c** CD14+/LPS activated cells stimulated with 50 µM succinate secrete higher levels of IL-1β compared to CD14+/LPS activated cells w/o succinate, $n = 4$ individual biological replicates, t-test, stim stimulated, Suc succinate. **d** Tube formation in hHUVECS (human umbilical vein endothelial cells) is increased upon succinate stimulation (50 µM), when added to conditions containing angiogenic growth factors and in growth factor free conditions, $n = 3$ biological replicates, Two-way ANOVA, Suc succinate. **e** Light microscopic pictures of tube formation after 18 h (+GF + FCS) showing representative pictures of control (w/o succinate), 50 µM, and 500 µM succinate, including images of identified tubular structures by ImageJ used to calculate tube length. GF growth factors, FCS fetal calf serum. **f** Repopulated area/scratch wound healing (%) by hMSCs is enhanced when succinate is added to the culture, in three independent donors after 24 h. $n = 3$-4 individual experimental replicates, One-way ANOVA. **g** Light microscopic images at 0, 12, and 24 h after scratch of treatment conditions (control, 50 µM, and 500 µM succinate). The scratch/cell layer boarder is indicated by the black line. **h** Analysis of hMSC migration rate every 3 h during the course of the 24 h-experiments. Addition of succinate enhances the migratory rate from early timepoints on. Pool of all three donors, *significant 50 µM to control, #significant 500 µM to control, Two-way ANOVA, CTRL control, Suc succinate, hMSCs human mesenchymal stromal cells. All graphs show mean ± standard deviation.

best of our knowledge, for the first time a study that analyzes and compares the metabolic profile of during early bone healing phases in a preclinical in vivo model of functional endogenous versus biologically compromised healing. We further highlight, that specific metabolites – here succinate – can act as a metabolic communicator and thereby steer successful bone tissue regeneration.

The presented metabolomic and proteomic analyses of regenerating bone hematoma and callus tissue revealed a time-dependent engagement of specific metabolic pathways that differ between the experimental groups of successful and compromised bone healing. Successful healing showed enhanced cellular metabolism within the complex hematoma and early callus tissue phases when compared to compromised bone healing. A higher anabolic demand in the group of successful endogenous healing may result in the accumulation of metabolites at metabolic check points of the CCM or upregulation of metabolic proteins. Upregulation of specific metabolites and metabolic proteins at different time points, in particular succinate and lactate at day 7, and at day 14, in parallel to proteins associated with oxidative phosphorylation, suggests a close connection between successful healing progression and local metabolic pathway engagement, which in contrast could not be seen in the group of compromised bone healing. The combination of metabolic and protein data (increased levels of glycolytic, TCA cycle and mitochondrial/oxidative phosphorylation proteins) complemented by gene expression analysis (higher levels of hexokinase 2 indicating increased shuttling of glucose into the CCM), gave us reason to assume that an enhanced cellular metabolism occurs across the complex hematoma and early callus tissue in successful healing (high anabolic demand) compared to compromised bone healing. The TCA-cycle intermediate succinate showed significant accumulation in fracture tissue of successful healing at day 7. However, the subsequently following TCA-cycle metabolites fumarate and malate, did not show such alterations, leading to the assumption that TCA cycle activity towards specific succinate formation or enrichment is reduced locally in compromised healing fracture tissue.

As mentioned before, significant work by Tannahill and colleagues, showed how intracellular accumulation of succinate affects *IL-1β* expression by HIF-1α stabilization in macrophages[27]. Within the fracture hematoma/callus environment macrophages and endothelial cells interact to promote angiogenesis, where HIF-1α plays a crucial factor[42]. Moreover, bone healing progression depends on the cell-cell interaction of macrophages and mesenchymal stromal cells[43,44]. We believe that altered cytokine expression of macrophages, e.g., by intracellular succinate accumulation, can influence this crosstalk and lead to healing cascade alterations. However, our experimental design with a usage of whole hematoma/callus tissue samples for measurements and analyses, did not permit an assessment of the cellular localization of the accumulated succinate.

Most records reporting on the function of succinate are either investigating the effect of intracellular or extracellular succinate. Several studies have highlighted how accumulation of intracellular succinate stabilizes HIF-1α leading to the expression of angiogenic and pro-inflammatory genes in inflammatory innate immune cells, such as macrophages and dendritic cells[27,35,45]. Extracellular succinate is also relevant for inflammatory processes and many studies have explored the pro-inflammatory potential of high levels of succinate on inflammatory macrophages, which contributes to disease aggravation e.g., in rheumatoid arthritis or cancer[27,29,35,46,47]. Succinate, however, can also mediate anti-inflammation, as recent studies show that macrophage derived-extracellular succinate suppresses inflammation in neurons and boost anti-inflammatory responses in adipose tissue[30,48,49]. These effects are mediated by SUCNR1 ligation[16,28], which is highly expressed in bone marrow, MSCs, and blood cells, particular monocytes and macrophages (Human protein atlas, Version 19.3)[30,31,50].

What is reported in the literature underlines the here presented finding of succinate acting as an extracellular modulator of cell function, here of human macrophages, endothelial cells, and MSCs. Taken together, these findings support the possibility of succinate as a metabolic communicator in bone healing and possibly other healing scenarios[28,48,49]. Two recent studies are in line with this hypothesis. While Mu and colleagues focused on succinate/succinate receptor 1 effects in promoting angiogenesis in cancer and used higher concentrations of succinate[36], it still demonstrates – together with our work – the potential of succinate driving angiogenesis in tissue regeneration. Ko et. al. showed a direct link of succinate, the succinate receptor 1, and tissue healing. They treated umbilical cord blood-derived stromal cells with 50 µM succinate before transplantation into a skin incision in mice. Wounds showed a faster closure when treated with succinate stimulated cells compared to wounds treated with cells where SUCNR1 signaling was blocked[38]. This is concurrent with our results of primary hMSCs obtained from the iliac crest of hip replacement patients, where addition of succinate increased hMSCs migration. Importantly, we additionally demonstrated that exposure to succinate increased osteogenic differentiation of hMSC, highlighting the great potential of metabolic communication and modulation for bone tissue regenerative approaches by acting presumably not only as a migratory stimulant for hMSCs within the fracture gap but also supporting osteogenesis and bone formation.

Interestingly, we found that already physiological concentrations of extracellular succinate let to an increased expression and secretion of a pro-inflammatory cytokine—IL-1β. As macrophages and MSCs work in close collaboration during bone healing[4,37,43] and succinate peaked at day 7 in the group of successful healing, we explored possible functions of IL-1β on bone healing progression.

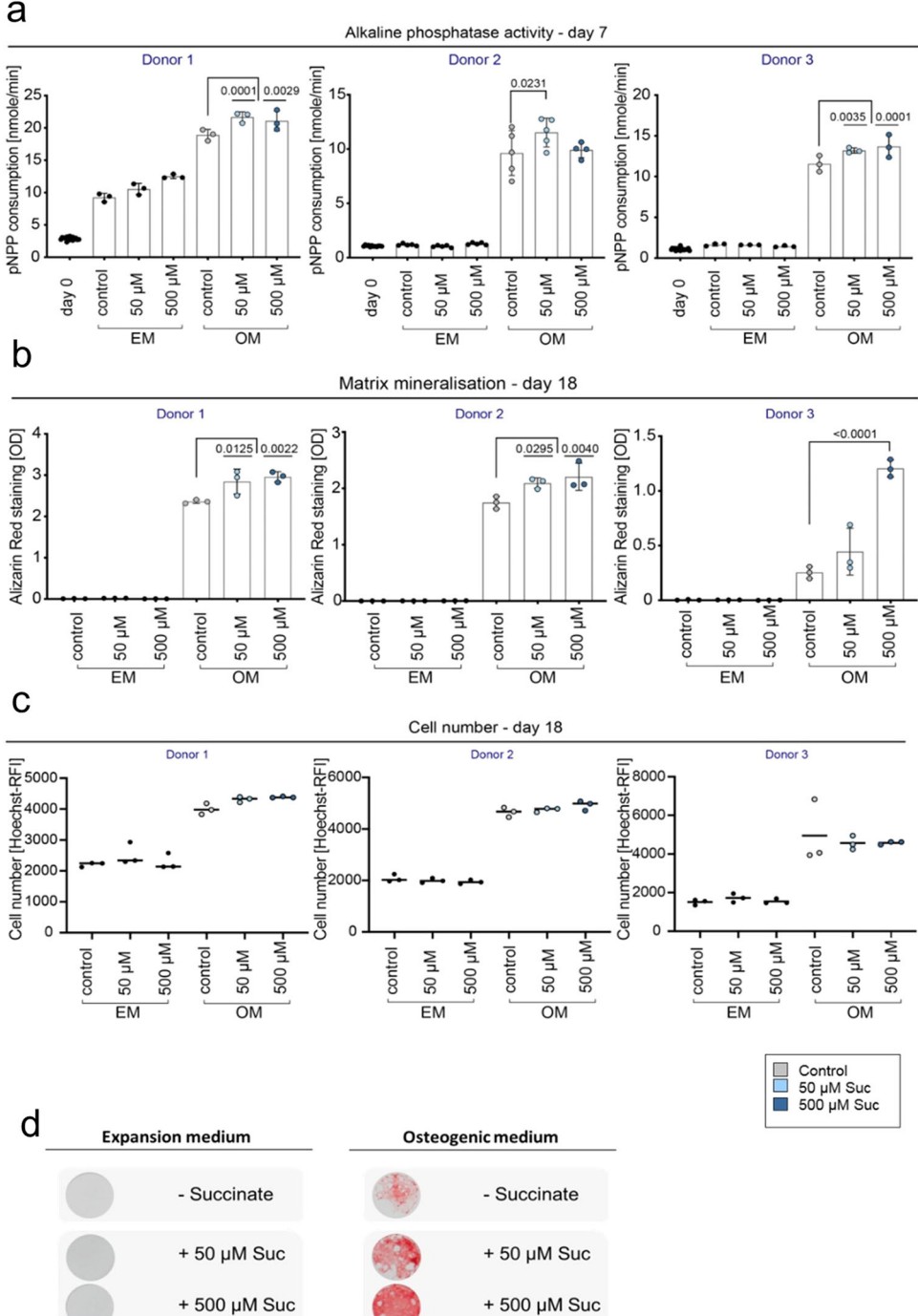

**Fig. 8 In vitro osteogenic differentiation of human MSCs is enhanced upon extracellular stimulation with succinate. a** Enzymatic activity of alkaline phosphatase, early phase marker of osteogenic differentiation, is significantly increased at day 7 in conditions with succinate treatment (50 μM, 500 μM), n = 3-5 individual experimental replicates per donor, One-way ANOVA, mean ± standard deviation. **b** Alizarin red staining used to identify mineralized matrix formation and osteogenic differentiation shows higher values in succinate treated conditions, compared to control osteogenic differentiation at day 18. n = 3 individual experimental replicates per donor, One-way ANOVA, mean ± standard deviation **c** Cell number determined by Hoechst staining is not altered between the treatments (control versus succinate) at day 18 of the differentiation. n = 3 experimental replicates per donor, dot plot shows the mean, One-way ANOVA. **d** Exemplary light microscopic picture of alizarin red staining at day 18 shown for osteogenic versus control (expansion medium) conditions and succinate treated wells. EM expansion medium, OM osteogenic medium, Suc succinate, pNPP 4-nitrophenylphosphate, OD optical density, RFI relative fluorescence intensity.

While Mumme et al. showed that constant IL-1β induced osteogenic and chondrogenic differentiation in vitro[39], our data – however – may point in a different direction. Continuous exposure of IL-1β to osteogenic cultures of hMSCs let to a diminished osteogenic differentiation. In contrast, when IL-1β was given only

for a specific time at the beginning of differentiation, it highly enhanced osteogenic differentiation of hMSCs. Around day 7 of successful bone healing in the here used animal model the fracture hematoma transitions towards callus formation, with chondrogenic areas forming that later develop into newly formed mineralized

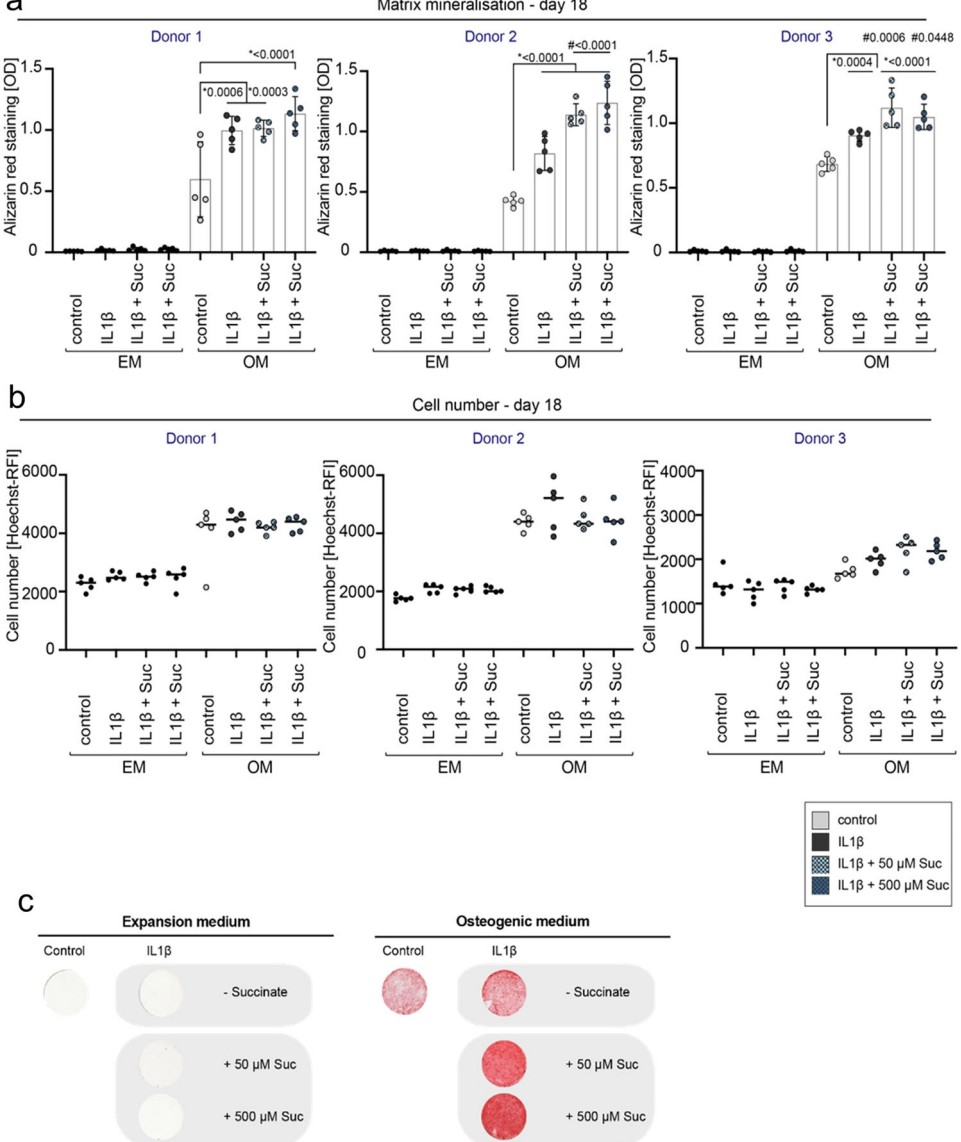

**Fig. 9 Combination of transient IL-1β and succinate stimulation during osteogenic differentiation of human MSCs strongly increases osteogenic differentiation potential in vitro. a** Addition of IL-1β (0.1 ng/ml) for 24 h after the induction of osteogenic differentiation resulted in increased levels of mineralized matrix compared to control osteogenic conditions (OM without succinate) after 18 days. Simultaneous addition of succinate (50 μM, 500 μM) to IL-1 β conditions, let to a further increase in mineralized matrix levels. $n = 5$ individual experimental replicates per donor, *significant to OM control, # significant to OM + IL-1β treatment, by One-way ANOVA. Shown are mean ± standard deviation. **b** Cell numbers as determined by Hoechst staining is not altered between the treatments (control versus IL-1β versus succinate) at day 18 of differentiation. $N = 5$ individual experimental replicates per donor, dot plot shows the mean, One-way ANOVA. **c** Exemplary light microscopic pictures of alizarin red staining at day 18 shown for osteogenic conditions versus non-osteogenic control conditions and IL-1β ± succinate treated wells. EM expansion medium, OM osteogenic medium, IL-1β interleukin 1 beta, Suc succinate, OD optical density, RFI relative fluorescence intensity.

tissue. Our findings and supporting reports from literature[51,52] indicate that the presence of IL-1β, possibly derived from macrophages stimulated by succinate, could be a factor steering mineralization at this phase of regeneration. Succinate, not only functions as an inducer of IL-1β but afterwards as a stimulator of osteogenic differentiation, which further potentiates mineralization. Conversely, prolonged IL-1β could lead to inflammatory signaling and result in compromising healing, in line with growing knowledge of successful endogenous bone healing being a fine-tuned process, where any alteration may lead to delays. First-line experiments in support of this hypothesis, have been performed by us by adding conditioned media from M1- and M2-like macrophages to osteogenic differentiation assay of primary human MSCs, thus

mimicking paracrine signals possibly received by hMSCs and secreted by macrophages during fracture healing. M2-like macrophages-related signals not only increased in vitro mineralization in osteogenic cultures compared to osteogenic control conditions but had M2-like macrophages been stimulated with extracellular succinate during polarization tissue mineralization was even more enhanced. Contrary, M1-like macrophage-related signals reduced in vitro mineralization in osteogenic conditions, more so when succinate was present during macrophage polarization.

Most interesting, paracrine signals from macrophages—whether M1-like or M2-like – stimulated in vitro mineralization of hMSCs, in the absence of osteogenic additives. Signals from M2-like macrophages, stimulated with succinate rendered the highest

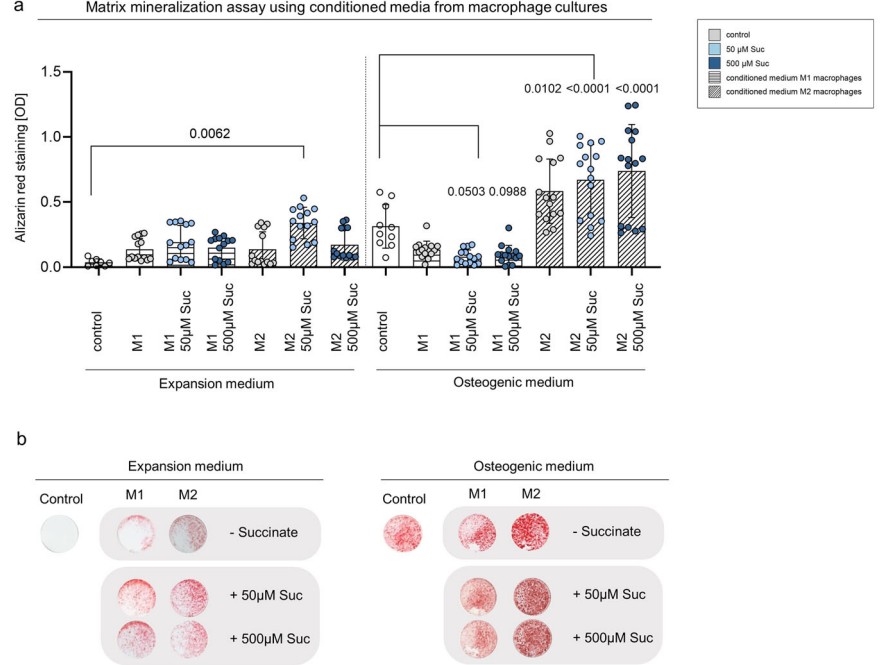

**Fig. 10 In vitro osteogenic differentiation of human MSCs is highly influenced by paracrine signals from conditioned medium from macrophage polarization with additive effects from succinate. a** Identified mineralized matrix formation by alizarin red staining shows higher levels in osteogenic conditions when media from M2-like cultures was added to the wells. This effect was higher when M2-like macrophages were stimulated with extracellular succinate during polarization. Media from M1-like macrophage cultures showed a decreasing effect on mineralization when compared to osteogenic control conditions. Media from macrophages stimulated mineralization in control conditions (EM, expansion medium) in the absence of osteogenic differentiation factors. EM expansion medium, OM osteogenic medium, OD optical density, Suc- succinate. M1- M1-like macrophages, M2-M2-like macrophages. Pool of 3 biological replicates/donors, $n = 3$ individual experimental replicates for each condition for each donor, One-way ANOVA. Shown are mean ± SD. **b** Exemplary light microscopic picture of alizarin red staining at day 14 shown for osteogenic versus control (expansion medium) conditions and media from macrophages ± succinate treated wells. EM expansion medium, OM osteogenic medium, Suc succinate. OD optical density, M1- M1-like macrophages, M2-M2-like macrophages.

levels of mineralization, suggesting a close link and fine-tuned interplay of different types of macrophages with MSCs and respective paracrine signals.

This study gives a first glimpse into the complex but essential role of the local metabolic microenvironment in enabling successful endogenous bone healing. However, several limitations need to be mentioned: As discussed earlier, we were not able to unravel the cellular origin of the increased succinate due to technical limitations. Therefore, all further investigations on cellular crosstalk and the involved cell populations within the local healing environment are still speculative. Whether the observed effects are indeed exclusively mediated over the succinate/SUCNR1-axis, a potential target for intervention and advanced therapies, is another important issue that needs to be investigated in further pre-clinical studies. Moreover, we could analyze healing characteristics earliest at day 3, since earlier specimen were still too fragile and instable for analysis, mainly consisting of the initial blood clot. This technical and practical limitation in turn means that we are unable to detect any significant differences in healing that might occur in the very early healing cascades. Besides metabolic targets, different patterns of extracellular matrix and adhesive molecules were identified by the proteomic analysis, indicating possible links between metabolism and extracellular matrix remodeling as additional future perspective. We believe nonetheless that tissue regenerative approaches can strongly benefit from the here presented and discussed findings concerning a metabolic regulation of bone healing and the postulated role of succinate as a potent metabolic cell-to-cell communicator.

We acknowledge once more, that our study marks only a starting point and more research is needed before this knowledge can be transferred towards therapeutic strategies. Especially single cell technologies, may enable to investigate the crosstalk of the distinct cell types in the fracture hematoma in the future.

## Methods

**Animal model.** In vivo animal studies were performed using 3- and 12-months old female ex-breeder Sprague Dawley rats purchased from Charles River WIGA Deutschland GmbH. As published, aged rats that had a minimum of three litters develop a fracture non-union after receiving a 2 mm osteotomy gap in the left femur if no further treatment is applied, therefore served as a model for biologically compromised bone healing in our study[19–21,25]. Four to six animals were randomly allocated to each group. All experiments were following ARRIVE guidelines, the National Institutes of Health Guide for the Care and Use of Laboratory Animals, and the National Animal Welfare Guidelines. Animal experiments were approved by the local legal representative (Institutional Animal Care and Use Committees, LaGeSo, G0120/14, G0172/15).

**Surgical procedure.** In order to assess bone healing a 2 mm femoral osteotomy was used as described and published[22–24]. In brief, prior to surgery animals were anesthetized by i.p. injection, antibiotics were given by s.c. injection and eye balm applied (Supplementary Table 1). A longitudinal skin incision and blunt fascia dissection were made to expose the left femur. To stabilize the bone an in-house developed external fixator was mounted. For detailed information on the development of the fixator system please refer to previous publications from Preiniger et al. and Strube et al.[21–24], as well as the supplement. The 2 mm standardized double osteotomy was introduced into the femoral bone by sawing, gap reproducibility was ensured by using a sawing template. At the end of the procedure, opened muscle fascia and skin were closed; an anesthetic antagonist and post-surgical analgesia were given (Supplementary Table 1). For a more detailed description of the procedure, please refer to the supplementary information.

**4.3 Animal sacrifice and sample harvest**. Animals were sacrificed at days 3, 7 and 14 post-surgery, after anaesthesia (i.p. injection), by intracardiac injection of 7 ml potassium chloride solution (KCl, 1 M) inducing a cardiac arrest. The left femur was dissected and dislodged from the joints and surrounding muscle tissue carefully removed. Dependent on the down-stream analysis, specimens were handled differently as follows: (1) For RNA and mass spectrometry analyses— 2 mm of fracture tissue and adjacent tissue 1 mm proximal and distal (region of interest, ROI) were snap frozen in liquid nitrogen and homogenized subsequently. It was ensured that the tissue remained frozen during the whole homogenization procedure. The frozen and ground tissue samples were split in three different vials for (a) proteomics, (b) metabolomics and (c) RNA extraction. (2) For histological and radiographic analyses, femurs were fixed in 4% paraformaldehyde/PBS solution for 24 h at 4 °C. (3) Intracardiac blood was collected from all animals before cardiac arrest. Blood serum was obtained by centrifugation, aliquoted and stored at -80 °C until further use.

**μCT & radiographic analysis**. Following harvesting and fixation, bones were rinsed with water for 45 min at RT. Femurs were transferred to PBS solution and directly loaded on a sample holder. The scans were performed with a VivaCT 40 microCT (Scanco Medical AG, Brüttisellen, Switzerland) at an isotropic voxel size of 12.5 μm. A 0.5 mm aluminum filter was employed and an x-ray tube voltage of 70kVp and a current of 114 μA. Reconstruction was carried out with a modified Feldkamp algorithm using the Scanco reconstruction software. Ring artifact reduction and beam hardening correction were automatically applied.

**Histology & histomorphometrical analysis**. Histological assessment of bone healing was performed on 5 μm thick cyro-sections according to the Kawamoto's film method[53]. After fixation, femurs were transferred into ascending sucrose solutions for cryo-protecting purposes, before they were embedded in the same orientation (proximal right, distal left) in SCEM-Medium and frozen by immersion into cold n-Hexan (Sigma-Aldrich, MI, USA). Movat-Pentachrome staining was used to differentiate between mineralized and soft tissues in the fracture zone (ROI)[23]. Tissues are stained in the following color-code: mineralized bone—yellow/orange, collagen—yellow, cartilage—green/blue, osteoid—dark red, elastic fibers—orange/red and nuclei—blue/black. Pictures were taken with Zeiss Axioscope 40 Microscope, 10x objective and condenser, and Imaging AxioVision LE Software (Carl Zeiss, Germany). Quantification of tissue within the ROI was done using a semi-automated method on blinded sections in ImageJ (Version 1.44p).

**Protein and peptide extraction**. Frozen and ground rat fracture tissue samples that have been put aside for proteomic analysis (as described under 4.3 "Animal sacrifice and sample harvest") were resuspended in urea buffer (8 M urea, 100 mM Tris-HCL, pH 8.25, Sigma-Aldrich, Germany) and sonicated using a Bioruptor®. The homogenate was centrifuged for 10 min, 4 °C, and the supernatant collected. Total protein concentration was determined by BCA colorimetric assay, 100 μg per sample taken for protein digestion. Samples were digested in urea buffer and DTT (2 mM Sigma-Aldrich, MI, USA), reducing the disulfide bonds. Idoacetamide (11 mM, Sigma-Aldrich, MI, USA) was subsequently added, preventing a new disulfide bond formation. A double digest with LysC and trypsin was conducted at 30 °C, digestion stopped by adding trifluoroacetic acid. 18 μg peptide mixture were desalted on STAGE Tips, eluted, dried, and reconstituted in 15 μl 0.5% acetic acid-water solution[54].

**Untargeted proteomics**. Peptides were separated by reverse-phase chromatography on an in- house made 25 cm columns C18-Reprosil-Saphir (Dr. Maisch, inner diameter: 75 μm, particle diameter: 1.9 μm) using nanoflow HPLC system (Agilent 1200, Agilent Technologies, CA, USA), coupled directly via nano-electrospray ion source (Proxeon) to linear ion trap quadrupole (LTQ) Orbitrap Velos (Thermo Fisher Scientific, MA, USA). Mass spectra were acquired in a data-dependent analysis switch between survey MS scan (m/z 300–1700, resolution $R = 60'000$) and MS/MS spectra acquisition. The 20 most intense ions (Top20) of each survey MS scan were selected for fragmentation and MS/MS spectra acquisition. Monocharged ions, potential contaminants, were excluded from analysis.

**Proteomic data analysis**. Raw files from the LTQ Exactive was processed using the MaxQuant computational proteomics pipeline and the built in peptide search engine Andromeda[55,56], species specific databases were loaded (UniProt rattus norvegius & common contaminants). An untargeted approach was performed, with LFQ settings (label-free quantifications) and unique peptide quantification to ensure isoform-specific calculation[57]. Other settings included: trypsin as a protease, with cleavage after lysine and arginine (restriction after proline), variable mod-ifications of methionine oxidation and N-terminal acetylation were chosen, and the peptide tolerance was set to 7ppm. Data quality was evaluated before further analysis by using an in-house developed software (PTXQC)[58]. Afterwards, data clean-up to remove contaminants and perform data normalization the Perseus software was used[59], the general workflow is depicted in Supplementary fig 6. To evaluate under- and over-represented protein groups between successful and compromised bone healing, the fold change was calculated. Values with a fold change <0.7 were considered underrepresented in compromised healing, while a

fold change of >1.6 was considered overrepresented/ enriched in compromised healing compared to successful bone healing.

**Metabolite extraction**. Metabolites were extracted from whole fracture hema-toma/callus tissue and blood serum as described before[60], with some modifications as described in the following. Frozen fracture tissue was homogenized as described under 4.3 animal sacrifice and weight. Per 50 mg frozen tissue 1 ml of methanol-chloroform-water (MCW; 5:2:1 v/v/v, Merck, Germany) was added, supplemented with 2 μg/ml cinnamic (Sigma-Aldrich, MI, USA) acid serving as internal standard. MCW-sample solution was further homogenized by ultra-sonication for twice for 30 s. Samples were shaken for 30 min at 10 °C, 200 rpm, water was added and shaken again for 5 min before centrifuged for 10 min at 10,000 x *g*, 4 °C to separate the polar metabolites from the lipid metabolites/phase. The aqueous, polar phase was collected and vacuum dried. For derivatization, 20 μl methoxyamine hydro-chloride solution (40 mg/ml in pyrimidine) (Sigma-Aldrich, MI, USA) was added to the dried fracture tissue extracts and incubated for 90 min at 30 °C. Next, 80 μl of N-methyl-N-(trimethylsilyl)trifluoroacetamide (MSTFA, VWR, PA, USA), including a retention index mixture (nine alcanes) was added to the mixture and incubated for 60 min at 37 °C, while shaking constantly. Mixture was centrifuged for 5 min at maximum speed and supernatant was split and transferred into glass vials for GC-MS measurement.

**Untargeted metabolomics**. All samples were measured by GC-TOF-MS as described[60,61]. In brief, samples were measured in a 1:5 split with 1 μl injection volume on a gas chromatography coupled to time of flight mass spectrometer (Pegasus III- time-of-flight (TOF)-MS-System, St. Joseph, MI, USA) from LECO®. Quality of injection and alkane peaks were regularly monitored, and temperature regulated. Temperature program during sample injection: 30 s, 80 °C, followed by a gradient with 12 °C/min up to 120 °C. Second ramp with 7 °C/min up to 300 °C and held for 2 min. Gas chromatographic separation was performed on an Agilent 6890 N, equipped with a VF 5 MS column and helium as a carrier gas at a flow rate of 1.2 ml/min. Temperature program after sample injection: 2 min, to 67.5 °C, temperature ramp up to 120 °C with 5 °C steps per min. Second ramp with 7 °C rise per min to 200 °C, followed by 12 °C per min to 320 °C, held for 6 minutes. Sample order was randomized, artefacts due to a longer standing time in experimental groups was thus avoided. Mass spectra were recorded in a mass range of 60 to 600 mass units with 10 spectra/s at a detector voltage of 1650 V.

**Metabolite data analysis**. The raw data and peaklists were extracted, baseline corrected, and resampled using the ChromaTOF® software. Data was subsequently read into the in-house analysis software MAUI-VIA for annotation, normalization, and quantification of metabolites by targeted library search for metabolite spectra. For detailed information about the procedure and the software please refer to Kuich et al.[26]

**Cell culture THP-1 and MP differentiation/polarization and primary CD14+ isolation**. Effects of extracellular succinate (50 μM and 500 μM) were studied on M1 and M2-like macrophages differentiated and polarized from the human monocytic cell line THP-1 as described before[62]. THP-1 (ECACC, Sigma-Aldrich/Merck) monocytes were seeded at $5 \times 10^5$–$1 \times 10^6$ cells per 6 well and stimulated for 72 h with 100 ng/ml Phorbol 12-myristate 13-acetate (PMA) in 1640-RPMI (Sigma-Aldrich, MI, USA) to induce adherence. They were put on cytokine and FCS-free 1640-RPMI media for 24 h before differentiated into M1 macrophages by 100 ng/ml LPS (Sigma-Aldrich, MI, USA) and 20 ng/ml IFNγ (Biolegend, CA, USA) and into M2 macrophages by using 20 ng/ml IL-4 (Miltenyi, Germany) and 10 ng/ml IL-13 (Miltenyi, Germany) for 72 h. Polarization state and transcriptional changes upon succinate stimulation were confirmed by qPCR. Primary human CD14 + cells were freshly isolated from human blood (commercially acquired buffy coats) by density-gradient isolation and MACs (Miltenyi, Germany) and seeded at $1 \times 10^6$ cells per well before stimulated with LPS and succinate (Sigma-Aldrich, MI, USA). IL-1β secretion was analyzed in the culture media after 24 h by human-Interleukin 1β Quantikine ELISA (R&D Systems, MN, USA).

**Gene expression, qPCR analysis**. For gene expression analysis of fracture tissue samples were harvested as described under 4.3 animal sacrifice. TRIzol Reagent (LifeTechnologies, CA, USA) was added to the homogenized tissue, and RNA isolated according to the manufacturer's protocol. For cell culture gene expression analysis, cells were lysed in RLT buffer and RNA isolated with RNeasy® Mini Kit (Qiagen, Netherlands) according to the manufacturer's protocol. RNA con-centration was determined using a Nano-Drop spectrophotometer. 25 ng/μl RNA was transcribed to cDNA with iScript reverse transcriptase as suggested by the manufacturer (Bio-Rad Laboratories, CA, USA). Gene transcript expression was determined by qPCR (LightCycler®480, Roche, Switzerland). A list of all genes tested, and the primer sequences can be found in Supplementary Table 2. Primer sequences were generated and tested for specificity using the NCBI website Primer-BLAST. Gene expression was analyzed according to the ddCT method with respective adjustment to primer efficiency and normalization to housekeeping genes by using the REST software[63]. Rat: the housekeeping gene Tata-box binding protein (*Tbp*) was tested against others (*Gapdh, Actb, Eif4e,* and *B2m*) and found

the most stable across all samples. Human: the housekeeping gene *RPL13A* was found the most stable-expressed in all sample and against other housekeeping genes tested (*TBP, ACTB, EIF4e* and *RLP0*) and therefore used as a reference gene.

**Tube formation assay.** HUVECs (bought: Lonza Switzerland, Lot 220213, 80016, 482213) were cultivated in endothelial growth medium (EGM, EGM BulletKit, Lonza, Switzerland) until 80% confluence. Tube formation was assessed as published[64]. In brief, $5 \times 10^4$ cells were plated per 24 pretreated/coated well with 50 µl growth factor reduced Matrigel (BD Biosciences. NJ, USA). To assess the effect extracellular succinate has on tube formation, cells were cultivated in EGM with growth factors, penicillin/streptomycin, and 2% FCS, ± succinate: 50 µM and 500 µM (+/+), in EGM without growth factors, supplemented with penicillin/streptomycin and 2% FCS ± succinate: 50 µM and 500 µM (−/+) and in EGM without growth factors, without 2% FCS but supplemented with penicillin/streptomycin ± succinate: 50 µM and 500 µM (−/−). Tube formation was documented after 18 h of culture by bright field microscopy. Total tube formation length per well was measured using ImageJ (version 1.44; http://rsbweb.nih.gov/ij/) and effect of succinate compared to control for each condition.

**Migration assay.** The migratory behavior of primary human MSCs upon extracellular succinate stimulation was assessed by using a scratch wound healing assay. $1 \times 10^5$ MSCs were seeded per 24 well and allowed to attach to a confluent layer overnight. Using a 200 µl pipette tip a scratch disrupting the cell layer was created. Cells were washed with PBS and media supplemented with penicillin/streptomycin and ±50/500 µM succinate but without FCS was added to the respective wells. Cell migration was tracked for 24 h, with a picture taken every 30 min using a bright field microscope (inverted DMI600B, Leica, Germany) with a life cell imaging system. Migration was analyzed using TScratch software[65] in a blinded approach for the different groups.

**Osteogenic differentiation of primary human MSCs.** Primary human MSCs were isolated from patient bone marrow, which was obtained during surgical procedures of hip or joint replacement, according to the ethics approval EA099/10 and after written informed consent once contagious maladies were excluded. To that end, bone marrow mononuclear cells were isolated by density gradient centrifugation and plastic adherence according to Pittenger et al.[66] Cells were expanded in DMEM for 14 days, then split and reseeded as passage 1. Cells were routinely checked for the prerequisites for human MSCs according to the International Society for Cellular Therapy[67,68]. For all experiments, MSCs in passage 3 were used.

Osteogenic differentiation was conducted was adapted after Krause et al. and performed as described before[66]. In brief, $2.05 \times 10^3$ cells were seeded per well (96-well plate) and were allowed to attach and form a confluent layer over night. Osteogenic media (DMEM low glucose, Gibco, NY, USA) was supplemented with 10% FCS (Biochrome, Germany), 1% Glutamax (Gibco, NY, USA), 1% penicillin/streptomycin (Biochrome, Germany), 0.1 µM dexamethasone (Sigma-Aldrich, MI, USA), 10 mM β-glycerolphosphate disodium salt hydrate (Sigma-Aldrich, MI, USA) and 50 µM L-ascorbic acid 2-phosphate sequimagnesium salt hydrate (Sigma-Aldrich, MI, USA) and additional substances as indicated (succinate, IL-1β). Control wells received normal MSCs expansion media (DMEM low glucose, 10% FCS, 1% Glutamax, 1% penicillin/streptomycin) and additional substances as indicated. Osteogenic differentiation was performed for a total of 18 days. Early osteogenic induction was assessed by alkaline phosphatase (ALP) activity at day 7, while late osteogenic differentiation was analyzed by matrix mineralization via alizarin red S staining at day 18.

Alkaline phosphatase activity was determined by colorimetric measurement. To that end, 4-nitrophenylphosphate was given to each well and the production of 4-nitrophenolate by ALP was determined by color intensity and optical density (OD) $\lambda = 405$ nm using a Tecan plate reader. Enzyme activity was calculated using the absorbance coefficient (ε). With ε = 18450 L x mol-1 x cm-1 and $d = 0.3294$ cm. C = $(E − E0)/\varepsilon \times d$, were E = absorbance (OD), E0 = absorbance blank (OD), c = molar concentration [mol/L], ε = molar absorbance coefficient [L x mol-1 x cm-1] and d = thickness of layer [cm]—e.g., 0.3294 cm for a 96-well, filled with 100 µL.

Matrix mineralization was visualized by alizarin red (AR) staining and quantified by colorimetry at OD of 562 nm. First, the cell number of each well was determined by Hoechst staining at $\lambda = 364$ nm/460 nm. Afterwards, wells were washed with H2O and overlaid with 0.5% alizarin red solution. Wells were washed 3 times with H2O to remove any excess or unbound staining solution. Last, the stained matrix was dissolved by adding cetylpyridnium chloride (Sigma-Aldrich, MI, USA) and determined by colorimetric analysis at $\lambda = 562$ nm. Treatment conditions (succinate, IL-1β) were compared to control/untreated wells.

**Osteogenic differentiation of primary human MSCs using conditioned media from macrophage cultures.** Osteogenic differentiation using conditioned medium from macrophage polarization cultures with and without the presence of succinate was performed as described under 4.16. Macrophage polarization medium was harvested, centrifuged, the supernatant aliquoted, and stored at −80 °C until use during osteogenic culture. The additives in the osteogenic differentiation medium were increased to a 2x concentration and mixed 1:2 with the conditioned medium from macrophage cultures respectively before adding it to the culture wells containing the hMSCs. Experiments were carried out with cells from three independent donors and each condition was tested in triplicates.

**ELISA.** Cytokine secretion from cell culture were measured using the human-Interleukin IL-1β Quantikine ELISA kit from R&D Systems, USA, MN (DLB50). Quantification was performed according to the manufacturer's instructions.

**Statistics and reproducibility.** Determined values are depicted as bar charts (with single values) or dot plots showing mean ± standard deviation. For statistical analysis GraphPad Prism 8.4. was used. All data were analyzed for normality distribution and subsequently tested with Welch's t-test or ANOVA using Tukey post-hoc correction for multiple testing. If normal distribution could not be confirmed within a data set, a non-parametric test was performed instead, either Man Whitney U Test or multiple pairwise comparison with Dunn's test. All test were performed as two-sided tests. P-values ≤ 0.05 were considered significant, trends are indicated by p-value in the respective figure. For each figure, the applied statistical method and amount of biological or experimental replicates are indicated in the figure legend.

## Data availability

The mass spectrometry proteomic datasets generated and analyzed during the current have been deposited to the ProteomeXchange Consortium (http://proteomecentral.proteomexchange.org) via the PRIDE[69] partner repository with the dataset identifier PXD020085. All other source data used for generating figures are provided as Supplementary Data files (Excel format).

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

## Acknowledgements

We are grateful for the excellent technical assistance of Janine Mikutta, Mario Thiele, and Jenny Grobe and for the excellent support of Christin Zasada and Guido Mastrubuoni regarding OMICs data acquisition and processing. We are also grateful for the provision of primary human MSCs from the BCRT Cell Harvest Core Facility (Simon Reinke). This work was made possible by funding for GND within the DFG CRC 1444, by support through a BIH translational Ph.D. project grant, and by the BCRT through funding by the German Federal Ministry of Education and Research (BMBF).

## Author contributions

J.L. has participated in the design and conception of the study, acquisition, analysis, and interpretation of data, manuscript writing and editing. A.N. has contributed to data analysis, interpretation of data, and manuscript editing. A.E. has participated in acquisition, analysis, and interpretation of data, and manuscript editing. A.D., G.N.D., and S.K. contributed to the conception and design of the study, interpretation of data, and manuscript editing. All co-authors approved the final version of the submitted manuscript.

## Funding

## Conflict of interests

No financial support or other benefits from commercial sources has been received by the authors for the work reported on in the manuscript.

## Competing interests

The authors declare no competing interests.
