## [Peer Review File · Communications Biology]

Reviewers' comments:

Reviewer #1 (Remarks to the Author):

This manuscript aims to provide a molecular profile of cells involved in bone fracture healing. The authors used an in vivo rat model of young (3 mo) vs old (12 mo) retired female breeders to compare successful fracture healing vs. compromised healing, respectively. They also follow these studies up with in vitro experiments. In sum, the authors conclude that the metabolite, succinate, is significantly lower during impaired fracture healing and provide evidence that it is specifically the attenuation of macrophage-succinate which drives impaired fracture healing in the aged mice via a mechanism involving IL-1 β . While this topic is of interest to a broad audience, the enthusiasm is slightly reduced due to major limitations/ weaknesses, and thus it is recommended that the manuscript include significant revisions prior to further consideration for publication.

Major weaknesses include:

- (1) Further details are needed about the sample used for proteomics. As this sample is expected to include a vast mixture of cells, these details are necessary. Furthermore, given this mixture, I am perplexed as to why the major conclusions drawn related to metabolic alterations are from macrophages. Why about chondrocytes, osteoblasts, osteoclasts, stromal cells, etc? The authors should try to focus on alterations occurring in purified cell populations.
- (2) Systemic metabolism is altered between young and aged rats; therefore, it is not unexpected that these analyses revealed such alterations. How do these alterations noted in the fracture callus compare to systemic changes? Can succinate be given back to aged rats to recover fracture healing like young rats? Is there a systemic issue or isolated to the fracture callus?
- (3) Authors should include validation data either by gene expression or protein abundance from the callus to investigate these pathways further.

In addition to these major weaknesses, minor weaknesses include:

- (1) Grammar throughout the manuscript should be checked.
- (2) It is important to note when the last litter was born from each retired breeder as this could profoundly impact fracture healing.
- (3) Include number of samples used for proteomics analyses in figure legend.
- (4) Figure 2 A and B is confusing. What is "underrepresented"? Are these the pathways/ proteins that are lower in the aged (compromised healing) models vs. young (successful healing)?

Reviewer #2 (Remarks to the Author):

The manuscript entitled "Metabolic alterations steer bone tissue regeneration - a comprehensive molecular profiling approach" by Löffler et al. is of high interest for the reader of Communications Biology.

The authors investigated the metabolic profile of successful and compromised bone repair in rats. For this, a model of young and aged female rats was used. They observed that central metabolic pathways were differentially regulated between the groups, which might contribute to the compromised healing outcome in aged rats. Furthermore, they identified in subsequent experiments that the metabolite succinate might play an important function in this process. With several in vitro experiments, the authors further confirmed that succinate fulfills several roles in, e.g., IL-1b induction in macrophages or osteogenic differentiation of MSCs, which have important functions during fracture healing. The study is highly relevant and very interesting and extends the current knowledge on metabolic demands during fracture healing. It is well written, and methods and results are clearly presented. The statistical analysis seems to be appropriate. However, some major points should be addressed for further clarification/correction. In particular, the results part includes already several literature references and "discussion" of results, which could be shortened in the results part and transferred to the discussion part.

Major points:

- What was the reason to focus on female rats? How can you exclude estrogen-dependent cyclic effects, which might influence your results? A short justification could be added.
- page 3/line 87: The biomechanics testing results you mention here are from previous studies, right? Why do you list this in your results part? Could be mentioned in the discussion. Furthermore, Fig. 1D is not showing the μ CT results, it is showing the histological results. Generally, there went something wrong with the numeration of figure 1. Also in the following, line 99, 102, 104, the numbers are not correct. Please check and correct.
- Fig. 1D and E. The standard deviation of the successful healing group on day 14 (mineralized tissue) is quite high. What is the reason for that? Do you observe differences in your fracture gap sizes that might influence with tissue regeneration capacity? Representative images of μ CT evaluations could be added to figure 1.
- Fig. 1F: Your later results show that inflammatory pathways are differentially regulated on protein level between the groups. Would be interesting to focus also on gene expression level on some inflammatory related genes, especially at time point day 3 after fracture. How many samples per group and time point were used for the gene expression analysis?
- It is interesting that succinate is differently expressed between the groups at day 7 after fracture, and that it is already known from the literature that succinate influences macrophages, which is why you further focussed on effects of succinate on macrophages in vitro. However, when are macrophages present during fracture healing and when do they peak? Is this corresponding to day 7? It should be more highlighted in the text. It would be interesting to know, if macrophage numbers are different at day 7 between successful and compromised bone healing in your rats? You could stain in older experiments macrophages in the fracture callus. Furthermore, it would be interesting if succinate associated pathways or receptors, especially SUCNR1, are differentially expressed at this time point in both groups and probably also in macrophages? These in vivo findings would further support your focus on macrophages in vitro.
- As osteoclasts derive from the same lineage as macrophages, it would be interesting to know, if metabolic profiles do also differ at later time points when callus is remodeled and osteoclasts are active. I know that you can not perform in vivo studies with a later time point within this study and time frame, however, it might be interesting to discuss this issue. Is something known about succinate and osteoclasts from the literature?
- As already mentioned, some of the literature and "discussions" from the results part should be transferred to the discussion part. And some limitations of the study should be included. For example, why not focussing on an earlier time point than day 3, as inflammation already starts very early after fracture and metabolic profiles might differ at earlier time points?
- page 14/line 367: You mentioned that not only the fracture hematoma, but also the adjacent tissue 1 mm distal and proximal was harvested for your analysis. How does this influence your results? You could critically discuss this point.

Minor points:

- Please introduce abbreviations properly (applies for the whole manuscript). E.g., CTSA, LAMP2,... Probably only few readers know these proteins, receptors, etc.
- page 6/line 154: you write that pyruvate and citrate levels were reduced, however, I do not see significant differences in citrate levels in Fig 3B (bar or p value of trend). Please check and correct if applicable.
- page 13/line 345: please also mention the group size somewhere here
- page 11/line 281: typo "inflammatory"
- page 14/line 357: stiffens of the fixation could be added
- page 16/417: typo "was" should be "were"

Point-by-point reply to

*“Metabolic alterations steer bone tissue regeneration – a comprehensive molecular profiling approach”
by Julia Löffler, Anne Noon, Agnes Ellinghaus, Anke Dienelt, Stefan Kempa and Georg N. Duda*

We would like to thank the editor and the reviewers for talking their time to read and critically comment our manuscript. We appreciated all comments and all aspects raised and tried to substantially improve the manuscript by revising it as well as by including additional data to support our claims. Please find below or point-by-point reply to the points raised by the reviewers. Thank you so much for your time and efforts!

Reviewer 1:

We thank reviewer 1 for the comprehensive and helpful feedback. We have addressed the issues raised point-by-point below.

Further details are needed about the sample used for proteomics. As this sample is expected to include a vast mixture of cells, these details are necessary. Furthermore, given this mixture, I am perplexed as to why the major conclusions drawn related to metabolic alterations are from macrophages. Why about chondrocytes, osteoblasts, osteoclasts, stromal cells, etc? The authors should try to focus on alterations occurring in purified cell populations.

Thank you for your feedback. Information about the samples used for proteomics have been specified in section 4.3 “Animal sacrifice and sample harvest” and described in more detail in section 4.6 “Protein and peptide extraction”. Additionally, the general sample workflow in Figure 1 has been updated, hoping to give a more concise image of our workflow.

Page 18, lines 460-464:

“(1) For RNA and mass spectrometry analyses – 2mm of fracture tissue and adjacent tissue 1mm proximal and distal (region of interest, ROI) were snap frozen in liquid nitrogen and homogenized subsequently. It was ensured that the tissue remained frozen during the whole homogenization procedure. The frozen and ground tissue samples were split in three different vials for (a) proteomics, (b) metabolomics and (c) RNA extraction.”

Page 19, lines 490-493:

“Frozen and ground rat fracture tissue samples that have been put aside for proteomic analysis (as described under 4.3 “Animal sacrifice and sample harvest”) were resuspended in urea buffer (8M urea, 100mM Tris-HCL, pH 8.25, Sigma-Aldrich, Germany) and sonicated using a Bioruptor®.”

We agree with the reviewer, that the samples used for analysis are a mixture of all cells involved in the fracture healing process at a given time. To our understanding, fracture healing is a highly orchestrated process of different biological phases, molecular signaling and cellular interactions as we and others have studied and published. ⁽¹⁻³⁾

However, the metabolic environment and its impact within this complex microenvironment has not yet been described. Here we aimed at giving a first glimpse into this highly dynamic topic, starting with giving

an insight into the overall metabolic profiles. Investigating individual metabolic profiles of purified cell populations is a significant step that should and will follow the here presented results.

The time-points of sampling have been chosen carefully, in order to reflect the significant stages of the fracture healing progress, as detailed in Figure 1B.

Within this manuscript, we focus particularly on the inflammatory phase (day 3), the switch from pro- to anti-inflammation and first angiogenic induction (day 7) and early reparatory phase, with a soft callus to early hard callus transition (day 14). By doing so, we aimed to construct the time sequence of successful endogenous and biologically compromised fracture healing at a local metabolic level. Especially the inflammatory phase and the angiogenic switch, are highly characterized by the presence of different types of macrophages, as published before by Schlundt et al. 2018 ⁽⁴⁾ and Löffler et al. 2019. ⁽⁵⁾ After metabolic analysis of the fracture tissue had been performed, we decided to further investigate macrophages in this context and focus on the possible impact metabolites can have on their functionality. It was not our intention to overestimate the role of macrophages during the fracture healing process and we are grateful for the advice. We have adjusted the aim of the manuscript and specific results sections, accordingly. We hope with such focus to convey this aspect in a more precise and understandable manner.

Page 7, lines 178-207:

“Particularly, a study by Tannahill and colleagues, demonstrated significant effects of intracellular accumulation of succinate on interleukin 1-beta (IL-1β) expression by hypoxia-inducible factor 1-alpha (HIF-1α) stabilization in macrophages.(27) Further studies showed that succinate has additional properties as an extracellular signaling molecule, mediated over its G-protein-coupled receptor, succinate receptor 1(SUCNR1). (16, 28, 29) Interestingly, SUCNR1 is highly expressed in bone marrow and blood cells, particularly MSCs, monocytes and macrophages (Human protein atlas, Version 19.3).(30, 31)

Gene expression analysis of the succinate receptor in fracture samples showed an increased expression in samples of successful healing at day 7 (Figure 4C). We were interested whether altered cytokine expression of macrophages, e.g., by intracellular succinate accumulation, may influence cellular crosstalk and lead to healing cascade alterations. Metabolic analysis of serum samples collected at the time of animal sacrifice confirmed that there was no systemic regulation of succinate between healing time points in successful or compromised healing (Supplementary figure 3). Unfortunately, localization and cellular origin of the accumulated succinate was not possible in the applied setting, due to the usage of homogenized hematoma/callus tissue samples.

We therefore decided to focus our attention on the potential function of extracellular succinate as a signaling molecule during the process of bone healing. We could demonstrate a differential expression of macrophage markers between young and old animals in a previous study, showing not only higher monocyte-macrophage marker expression, like CD14 and CD68 in fracture tissue of successful healing but also increased anti-inflammatory M2 macrophage gene expression.(23) Additionally, angiogenic marker genes, like Hif-1alpha and others associated with overall vessel formation were increased in fractures from young animals at day 7. Further studies performed at our institute confirmed that M2 macrophages appear in the fracture gap at around day 7 in the selected animal model.(32) We assumed that the observed difference in succinate levels between the successful and compromised healing animals could be related to macrophages and aimed to explore this potential crosstalk further.

We did so, by simulating a local exposure of succinate to cells relevant in bone healing, like macrophages, mesenchymal stromal cells (MSCs) or endothelial cells in vitro, aiming to mimic the microenvironment during successful fracture healing.”

We agree with the reviewer about the benefits of investigating purified cell populations from fracture tissue. Unfortunately, we were – in this study – confronted with some technical difficulties like the availability of specific antibodies and sufficient sample material. A more detailed cellular analysis beyond macrophages is, however, now ongoing and focus of current pre-clinical and clinical investigations.

In the here presented manuscript, sufficient cellular material would only have been reached by extracting cell populations from multiple animals and be pooled. This was, however, beyond the capabilities and scope of the animal approval and thus not allowed herein due to ethical considerations.

For a more in-depth characterization of these cell populations, multi-marker-analyses by e.g. flow cytometry would be needed prior to pooling samples. Bone callus tissue is a very diverse and very dynamic tissue. It transfers from hematoma/ fibrin rich clotted blood, bone marrow and granulation tissue to bone, cartilage, and fibrous tissue. To the best of our knowledge, there is no yet an established protocol that allows to dissect these diverse cell types in a proper style and still consider their metabolomic profile (despite the processing) to reflect the in vivo status, both qualitative and quantitative. However, some groups – including us - are working on developing such technologies to enable a more sophisticated multi-marker analyses of the diversity and dynamics of cell types.

For the current manuscript, we underwent substantial efforts trying to perform droplet-based single cell RNA-seq from whole hematoma samples with little success, as RNA seemed to be too unstable and subsequent cDNA library preparations continued to fail.

To that reason, we have herein focused on giving a first overview from whole hematoma/fracture tissue. We hope that the reviewer agrees with this procedure based on the information provided above. We hope to – in due course – be able to analyze more specific cell populations to generate valuable insights in cell specific metabolomic dynamics during such complex processes as bone healing. To be able to do so, we will in the future switch to fracture healing in mice and aim at applying 10x- single cell genomics.

Systemic metabolism is altered between young and aged rats; therefore, it is not unexpected that these analyses revealed such alterations. How do these alterations noted in the fracture callus compare to systemic changes? Can succinate be given back to aged rats to recover fracture healing like young rats? Is there a systemic issue or isolated to the fracture callus?

We thank the reviewer for the in depths questions about succinate rescuing experiments. We have performed a series of new experiments aiming to find further answers.

For insights about systemic metabolic changes, we performed untargeted metabolomic screening as described under 4.9. “Metabolite extraction” and 4.10 “Untargeted metabolomics” of systemic blood serum samples from animals sacrificed at day 3 and day 7 after osteotomy and both experimental groups (successful healing/young rats and compromised healing/aged rats. Systemic blood was collected for each animal by intracardiac blood collection before induction of cardiac arrest. Blood serum was obtained by centrifugation and stored at -80°C until further use, as now stated in the manuscript under section 4.3 “Animal sacrifice and sample harvest”

Page 18, lines 466-467:

“(3) Intracardiac blood was collected from all animals before cardiac arrest. Blood serum was obtained by centrifugation, aliquoted and stored at -80°C until further use.”

We have included figures of systemic metabolite levels in the supplement, figure 3 and adjusted the manuscript accordingly (see page 7lines 188-191). The metabolic levels observed at systemic level do not match with metabolic alterations seen in the fracture callus. We conclude that the metabolic alterations observed by us in the fracture callus are independent from the systemic metabolic profiles.

Performing succinate rescue experiments is of high significance to further evaluate the role of succinate in the fracture healing cascade. Beforehand, we would like to address some technical difficulties we were faced with, regarding *in vivo* rescue experiments. First, a systemic application of succinate would not result in a local accumulation of succinate within the fracture tissue, thus not provide us with the required conditions to properly answer this question. Second, local application of succinate, which would be better fitted for this purpose, requires a local injection in the fracture tissue, interfering with the local dynamics of the fracture cascade in an uncontrolled manner and lacking the ability to claim that this injection, succinate itself or its interaction within the local niche would be casual to changes observed.

To reach more relevance towards translation, we decided to use human MSCs obtained from elderly patients that underwent hip or joint replacement surgery in a first step as described in the manuscript in the results paragraph. Page 9 *"2.4 Extracellular succinate enhances angiogenic and migratory processes of primary human cells in vitro"* and page 10 *"2.5 Osteogenic differentiation potential of primary human MSCs increased by extracellular stimulation with succinate in vitro"*.

The stimulation by succinate did result in a faster and more profound mineralization in *in vitro* assays of osteogenic differentiation. This effect was most pronounced in hMSCs from donor 3, which showed less osteogenic potential compared to MSCs from donor 1 and 2, indicating that succinate not only increased mineralization during osteogenic differentiation of MSCs in general but gave an additional mineralization stimulus in cells showing less endogenous potential. This aspect has been specified in the manuscript as follows:

Page 10, lines 267-269:

"Addition of succinate (500 μ M) particularly induced strong mineralization, when the overall mineralization capacity appeared delayed or less pronounced, as suggested by observations from donor three (Figure 6C)."

Additionally, we performed experiments using conditioned media from macrophage differentiation assays with and without succinate addition and have added a completely new results section that elaborates on MSC and macrophage interaction further.

Page 11, lines 282-301

"2.7 Paracrine signals from macrophages stimulated by extracellular succinate generate osteogenic induction in the absence of further osteogenic stimuli in vitro"

To further investigate the paracrine crosstalk between succinate-stimulated macrophages and MSCs during bone healing progression, osteogenic differentiation assays were performed using conditioned media from M1- and M2-like macrophages that have been cultured and polarized with and without succinate (50 and 500 μ M) addition to the culture medium. This culture media was harvested and given to osteogenic differentiation assays using primary human MSCs. The addition of media from M2-like macrophages in conditions with osteogenic additives led to a significant increase of matrix mineralization as determined by alizarin red staining. This effect was even more pronounced when succinate was added to the cultures during macrophage polarization and stimulation. Addition of conditioned media from M1-like macrophages however, showed decreased mineralization, again the effect was stronger, when succinate was added to the cultures during macrophage polarization (Figure 8A, B).

Most astonishing was the effect observed in cultures without any osteogenic supplements, that usually served as control conditions and where expansion medium was used. Here addition of conditioned media from macrophage polarizations induced mineralization, independent of macrophage polarization (M1 or M2) as seen in Figure 8A and

8B. In control conditions, when media from M2-like macrophages stimulated with 50 μ M of succinate during polarization and differentiation was added, levels of mineralization showed the highest results.”

Authors should include validation data either by gene expression or protein abundance from the callus to investigate these pathways further.

Thank you for the mindful advice. Information on protein cluster and protein levels can be found in the manuscript, figure 3C. We decided to graph protein levels of the inflammatory cluster and metabolic cluster in a heat map, aiming to provide a significant amount of information in a simplistic way (Figure 2B). The heatmap was compiled using normalized label free quantification (LFQ) intensities, a standard approach to report on the relative amount of the respective protein in each sample. Additionally, we have provided extensive tables in the supplement reporting on the individual proteins identified for the differentially regulated cluster between successful and compromised healing. Exemplary bar graphs for protein abundance have been added to the supplement figure 1.

We have additionally performed gene expression analysis of selected inflammatory and metabolic genes supporting our finding from protein analysis. The data and respective graphs have been added to Figure 2D.

Gene expression analysis regarding macrophage abundance in fracture callus has been published by us previously and we would like to refer to the following publication for more information, Löffler et al, 2019⁽⁵⁾ and our answer to reviewer #2 page 8 of this point-by-point reply.

Minor weakness

1. Grammar throughout the manuscript should be checked.

We have assessed spelling and grammar thoroughly and hope to have made substantial improvements throughout the entire manuscript. Thank you for your critical review.

2. It is important to note when the last litter was born from each retired breeder as this could profoundly impact fracture healing.

The here applied model has been well characterized and used for different studies previously. Unfortunately, we cannot report the exact dates or time points of the last litter. However, our approximation is that the last litter was born around 4-6 weeks before delivery to our animal housing. By this time, the offspring is safely weaned. We ordered ex-breeder females aged between 7-8 months with at least 3 litters. Since the companies do not keep the animals after exclusion from breeding, we kept the animals for 4-5 months in our in-house facilities before surgical procedure. A more detailed section about the used *in vivo model* has been included in the supplement.

3. Include number of samples used for proteomics analyses in figure legend.

Thank you for the helpful comment – we apologize for missing the specific sample number in figure 2. We have included this information in the figure legend.

4. Figure 2 A and B is confusing. What is “underrepresented”? Are these the pathways/ proteins that are lower in the aged (compromised healing) models vs. young (successful healing)?

The terms we have chosen indeed aimed to highlight which proteins and protein cluster showed lower abundance and less enrichment in samples of aged animals/compromised healing. We are grateful for the remark and have rephrased “underrepresented” in the results part as well as figure 2. We hope to have made it less confusing and more concise by using “up-regulated” instead.

Reviewer 2

Thank you for your careful review and your suggestions. We will try to reply on a point-by-point basis below.

What was the reason to focus on female rats? How can you exclude estrogen-dependent cyclic effects, which might influence your results? A short justification could be added.

We would like to thank the reviewer for bringing up this point, giving us the opportunity to give more details about the chosen *in vivo* model. In several studies that have been performed and published by co-workers and colleagues, describe the biologically compromised/impaired fracture healing potential of aged female (twelve-month-old female ex-breeder Sprague-Dawley rats) that previously had a minimum of three litters.⁽⁵⁻¹⁰⁾ In order to exclude sex-differences in healing capacity we considered young female rats as a suitable control group, as has been done in previously published studies (Ref). We cannot exclude estrogen-dependent cyclic effects as a possible source of influence. However, since we have applied this experimental *in vivo* model for many previous studies and were able to publish consistent results using it, we are confident that it is suitable to mimic the clinical problem of age and sex related fracture healing in female patients.

We have adjusted the respective section in the material and method section 4.1 “Animal model” and 4.2 “Surgical procedure” trying to clarify the background and purpose of the used model. Additionally, we have included a section in the supplement, elaborating in more detail on the *in vivo* model.

Page 17, lines 434-439:

“In vivo animal studies were performed using 3- and 12-months old female ex-breeder Sprague Dawley rats purchased from Charles River WIGA Deutschland GmbH. As published, aged rats that had a minimum of three litters develop a fracture non-union after receiving a 2 mm osteotomy gap in the left femur if no further treatment is applied, therefore served as a model for biologically compromised bone healing in our study (19-21, 25). Four to six animals were randomly allocated to each group.”

Page 17, lines 447-452:

“A longitudinal skin incision and blunt fascia dissection were made to expose the left femur. To stabilize the bone an in-house developed external fixator was mounted. For detailed information on the development of the fixator system please refer to previous publications from Preiniger et al. and Strube et al.(21-24), as well as the supplement The 2 mm standardized double osteotomy was introduced into the femoral bone by sawing, gap reproducibility was ensured by using a sawing template”

page 3/line 87: The biomechanics testing results you mention here are from previous studies, right? Why do you list this in your results part? Could be mentioned in the discussion. Furthermore, Fig. 1D is not showing the μ CT results, it is showing the histological results. Generally, there went something wrong with the numeration of figure 1. Also in the following, line 99, 102, 104, the numbers are not correct. Please check and correct.

We would like to thank you for the critical review and apologize for inconsistencies in figure numbers. We went through the text and corrected the figure numeration. During the revision we have split figure 1 und now two different figures, where figure 1 gives details on the here selected model and previous results, while figure 2 now shows validation data for early healing timepoints only.

Fig. 1D and E. The standard deviation of the successful healing group on day 14 (mineralized tissue) is quite high. What is the reason for that? Do you observe differences in your fracture gap sizes that might influence with tissue regeneration capacity? Representative images of μ CT evaluations could be added to figure 1.

We thank you for your careful observation and would like to add some details about the femoral osteotomy that was performed on all animals. Animals received a 2mm osteotomy at the left femur, a sawing template was used to ensure a standardized procedure and bone defect. An in-house produced external fixator was mounted to the left femur before the osteotomy was created in the center between the two inner K-wires and removal of the bone segment. For further details about the surgical procedure, we kindly refer to previously published studies⁵⁻¹⁰ as well as the newly inserted detailed information in the supplement.

All animals and fixators were monitored daily and osteotomized femurs carefully dissected after animal sacrifice, with the external fixators still in place. All animals included in our analysis did not show differences in fracture gap size and we are confident that this parameter did not influence tissue regeneration capacities or outcomes. The here applied model of successful versus biologically compromised healing has been successfully characterized, used, and published. However, we are aware that in vivo models might differ slightly due to biological variance. Our personal experience and publication evidence makes us confident about the overall usability of the selected model.

Fig. 1F: Your later results show that inflammatory pathways are differentially regulated on protein level between the groups. Would be interesting to focus also on gene expression level on some inflammatory related genes, especially at time point day 3 after fracture. How many samples per group and time point were used for the gene expression analysis?

For gene expression analysis the same number of samples was used per time point and group as for protein and metabolic level analysis, as the homogenized fracture tissue was split in three parts as described in section 4.3 "Animal sacrifice". To make this procedure more understandable, the respective passage now states:

Page 18, lines 460-466:

"Dependent on the down-stream analysis, specimens were handled differently as follows: (1) For RNA and mass spectrometry analyses – 2mm of fracture tissue and adjacent tissue 1mm proximal and distal (region of interest, ROI) were snap frozen in liquid nitrogen and homogenized subsequently. It was ensured that the tissue remained frozen during the whole homogenization procedure. The frozen and ground tissue samples were split in three different vials for (a) proteomics, (b) metabolomics and (c) RNA extraction. (2) For histological and radiographic analyses, femurs were fixed in 4% paraformaldehyde/PBS solution for 24 h at 4°C."

The number of replicates for gene expression analysis is additionally specified in the figure legend 2D, and we apologize for not being precise enough.

We realized that a more detailed work-flow description is needed and therefore adjusted figure 1 B, now displaying a more comprehensive yet simplistic workflow. We thank the reviewer for raising this point and hope to have increased the clarity of our experimental set-up.

We agree with the reviewer and thank for raising the point regarding inflammatory gene expression validation. We have added expression data in Figure 3D and adjusted the manuscript accordingly.

Page 5, lines 138-145:

“Gene expression analysis of inflammatory markers and metabolic enzymes were performed to validate findings on protein level. Expression of pro-inflammatory genes like the tumor necrosis factor-alpha (TNF-alpha) or nitric oxide synthase 2 (Nos2) was high in compromised healing samples at day 3. In contrast, expression of the anti-inflammatory cytokine interleukin-10 (Il-10) was higher in fracture samples from successful healing at day 3 and 7. Similarly, central carbon metabolic enzymes (hexokinase 2 and succinate dehydrogenase beta) showed higher expression levels in successful healing samples during early healing time points, namely day 3 and day 7 (Figure 3D).”

It is interesting that succinate is differently expressed between the groups at day 7 after fracture, and that it is already known from the literature that succinate influences macrophages, which is why you further focused on effects of succinate on macrophages in vitro. However, when are macrophages present during fracture healing and when do they peak? Is this corresponding to day 7? It should be more highlighted in the text. It would be interesting to know, if macrophage numbers are different at day 7 between successful and compromised bone healing in your rats? You could stain in older experiments macrophages in the fracture callus. Furthermore, it would be interesting if succinate associated pathways or receptors, especially SUCNR1, are differentially expressed at this time point in both groups and probably also in macrophages? These in vivo findings would further support your focus on macrophages in vitro.

We would like to thank the reviewer for raising these questions and add some details to the theory behind the focus on macrophages. From previous studies in the institute, we know that macrophages are present from very early on in the fracture and their levels seem to peak at day 7.⁽⁴⁾ The shift from M1 to M2 macrophages is a strong indicator for the transformation of pro- to anti-inflammation in the early hematoma and a crucial step in successful bone healing progression.

Although M2 macrophages started to appear 3-7d post-surgery, the fracture area is pre-dominated by M2 macrophages at day 7. In a previous study we could show that M2 marker expression levels were significantly lower in bone callus tissue derived from aged animals compared to younger animals while inflammatory M1-macrophage marker were significantly higher in fractures from aged animals at day 3.⁽⁵⁾ This finding was supportive to us that macrophages are likely involved in the succinate regulation.

We have rewritten the passage in the results section 2.2 and hope to have clarified the focus on macrophages. Additionally, we have now included gene expression analysis of fracture samples for the succinate receptor 1.

Page 7, lines 181-207:

“Further studies showed that succinate has additional properties as an extracellular signaling molecule, mediated over its G-protein-coupled receptor, succinate receptor 1(SUCNR1). (16, 28, 29) Interestingly, SUCNR1 is highly expressed in bone marrow and blood cells, particularly MSCs, monocytes and macrophages (Human protein atlas, Version 19.3).(30, 31)

Gene expression analysis of the succinate receptor in fracture samples showed an increased expression in samples of successful healing at day 7 (Figure 4C). We were interested whether altered cytokine expression of macrophages, e.g., by intracellular succinate accumulation, may influence cellular crosstalk and lead to healing cascade alterations. Metabolic analysis of serum samples collected at the time of animal sacrifice confirmed that there was no systemic regulation of succinate between healing time points in successful or compromised healing (Supplementary figure 3). Unfortunately, localization and cellular origin of the accumulated succinate was not possible in the applied setting, due to the usage of homogenized hematoma/callus tissue samples.

We therefore, decided to focus our attention on the potential function of extracellular succinate as a signaling molecule during the process of bone healing. We could demonstrate a differential expression of macrophage markers between young and old animals in a previous study, showing not only higher monocyte-macrophage marker expression, like CD14 and CD68 in fracture tissue of successful healing but also increased anti-inflammatory M2 macrophage gene expression (23) Additionally, angiogenic marker genes, like Hif-1alpha and overall vessel formation was increased in fractures from young animals at day 7. Further studies performed at our institute further confirmed that M2 macrophages appear in the fracture gap at around day 7 in the selected animal model.(32) We assumed that the observed difference in succinate levels between the successful and compromised healing animals could be related to macrophages and aimed to explore this potential crosstalk further.

We did so, by simulating a local exposure of succinate to cells relevant in bone healing, like macrophages, mesenchymal stromal cells (MSCs) or endothelial cells in vitro, aiming to mimic the microenvironment during successful fracture healing.”

As osteoclasts derive from the same lineage as macrophages, it would be interesting to know, if metabolic profiles do also differ at later time points when callus is remodeled, and osteoclasts are active. I know that you cannot perform in vivo studies with a later time point within this study and time frame, however, it might be interesting to discuss this issue. Is something known about succinate and osteoclasts from the literature?

We would like to thank you for raising this interesting point. There is dual mechanism in play with succinate and osteoclasts. On one hand, succinate is known to promote osteoclastogenesis *in vitro* and *in vivo*. On the other hand, the accumulation of intracellular succinate induced by succinate dehydrogenase (SDH) inhibition is suppressing osteoclast formation. ⁽¹¹⁾

As already mentioned, some of the literature and "discussions" from the results part should be transferred to the discussion part. And some limitations of the study should be included. For example, why not focusing on an earlier time point than day 3, as inflammation already starts very early after fracture and metabolic profiles might differ at earlier time points?

The issues raised by the reviewer are highly appreciated and we have moved several aspects from the results part to the discussion part. Moreover, we have extended our discussion by illuminating study limitations. We are in full agreement on the interest of analyzing earlier time points (day 1 and 2), unrevealing the immediate inflammatory and immune response yet have been faced with technical and biological difficulties, making this impractical. Earliest bridging by some connective fibers appears around day 3 and any femur/fracture hematoma dissection tried at earlier time points resulted in the immediate dissolution and disintegration of the fracture hematoma. Even at day 3 all specimens need to be dissected and handled with the outmost of care and practice and are still very instable.

Page 16, lines 410-421:

“This study gives a first glimpse into the complex but essential role of the local metabolic microenvironment in enabling successful endogenous bone healing. However, several limitations need to be mentioned: As discussed earlier, we were not able to unravel the cellular origin of the increased succinate due to technical limitations. Therefore, all further investigations on cellular crosstalk and the involved cell populations within the local healing environment are still speculative. Whether the observed effects are indeed exclusively mediated over the succinate/SUCNR1-axis, a potential target for intervention and advanced therapies, is another important issue that needs to be investigated in further pre-clinical studies. Moreover, we could analyze healing characteristics earliest at day 3, since earlier specimen were still too fragile and instable for analysis, mainly consisting of the initial blood clot. This technical and practical limitation in turn means that we are unable to detect any significant differences in healing that might occur in the very early healing cascades.”

page 14/line 367: You mentioned that not only the fracture hematoma, but also the adjacent tissue 1 mm distal and proximal was harvested for your analysis. How does this influence your results? You could critically discuss this point.

By including adjacent tissue, located 1 mm distal and proximal to the fracture hematoma we aimed to also include cells and signals from the bone marrow that are attracted to the fracture site and thus involved in fracture healing cascades. This is especially relevant in later time points of the healing cascade, where relevant players migrate from the bone marrow cavity into the fracture gaps. In order to obtain tissue samples that can be compared in the best possible/standardized way this approach was chosen, we have adjusted the method section in the supplement accordingly.

Minor weakness

1. Please introduce abbreviations properly (applies for the whole manuscript). E.g., CTSA, LAMP2,... Probably only few readers know these proteins, receptors, etc.

Thank you for the advice. We now hope to have included all necessary information.

2. page 6/line 154: you write that pyruvate and citrate levels were reduced; however, I do not see significant differences in citrate levels in Fig 3B (bar or p value of trend). Please check and correct if applicable.

We apologize for the mistake made and have adjusted the manuscript accordingly. No differences in pyruvate or citrate levels were detected.

3. page 13/line 345: please also mention the group size somewhere here

We have added the animal group size in section 4.1 “Animal model” as requested, it now reads on page 17, lines 438-439 “Four to six animals were randomly allocated to each group”

4. page 11/line 281: typo "infammatory" - corrected and now reads “inflammatory”
5. page 14/line 357: stiffens of the fixation could be added

For the stiffness of the fixators we would like to refer to previous publication from Preiniger et al. and Strube et al.

6. page 16/417: typo "was" should be "were" - Thank you now corrected in the manuscript

We are grateful for the profound examination and have adapted and corrected the issues raised above in the manuscript. We feel that it has substantially improved our manuscript.

References

1. Schmidt-Bleek K, Schell H, Lienau J, Schulz N, Hoff P, Pfaff M, et al. Initial immune reaction and angiogenesis in bone healing. *J Tissue Eng Regen M.* 2014;8(2):120-30.
2. El-Jawhari JJ, Jones E, Giannoudis PV. The roles of immune cells in bone healing; what we know, do not know and future perspectives. *Injury.* 2016;47(11):2399-406.
3. Gerstenfeld LC, Cullinane DM, Barnes GL, Graves DT, Einhorn TA. Fracture healing as a post-natal developmental process: Molecular, spatial, and temporal aspects of its regulation. *J Cell Biochem.* 2003;88(5):873-84.
4. Schlundt C, El Khassawna T, Serra A, Dienelt A, Wendler S, Schell H, et al. Macrophages in bone fracture healing: Their essential role in endochondral ossification. *Bone.* 2018;106:78-89.
5. Löffler J, Sass FA, Filter S, Rose A, Ellinghaus A, Duda GN, et al. Compromised Bone Healing in Aged Rats Is Associated With Impaired M2 Macrophage Function. *Frontiers in Immunology.* 2019;10(2443).
6. Strube P, Mehta M, Baerenwaldt A, Trippens J, Wilson CJ, Ode A, et al. Sex-specific compromised bone healing in female rats might be associated with a decrease in mesenchymal stem cell quantity. *Bone.* 2009;45(6):1065-72.
7. Strube P, Mehta M, Putzier M, Matziolis G, Perka C, Duda GN. A new device to control mechanical environment in bone defect healing in rats. *J Biomech.* 2008;41(12):2696-702.
8. Strube P, Sentuerk U, Riha T, Kaspar K, Mueller M, Kasper G, et al. Influence of age and mechanical stability on bone defect healing: Age reverses mechanical effects. *Bone.* 2008;42(4):758-64.
9. Preininger B, Gerigk H, Bruckner J, Perka C, Schell H, Ellinghaus A, et al. An Experimental Setup to Evaluate Innovative Therapy Options for the Enhancement of Bone Healing Using Bmp as a Benchmark - a Pilot Study. *Eur Cells Mater.* 2012;23:262-72.
10. Mehta M, Duda GN, Perka C, Strube P. Influence of gender and fixation stability on bone defect healing in middle-aged rats: a pilot study. *Clinical orthopaedics and related research.* 2011;469(11):3102-10.
11. Guo Y, Xie C, Li X, Yang J, Yu T, Zhang R, et al. Succinate and its G-protein-coupled receptor stimulates osteoclastogenesis. *Nature Communications.* 2017;8(1):15621.

REVIEWERS' COMMENTS:

Reviewer #1 (Remarks to the Author):

The authors have addressed the original concerns and the manuscript is acceptable.

Reviewer #2 (Remarks to the Author):

The authors carefully reviewed the manuscript according to the comments. The manuscript significantly improved.